# LOOKING BEYOND THE NEXT TOKEN

## ABSTRACT

The most natural way to model language is rarely autoregressive. The structure of causal language model training assumes that each token can be predicted from prior context, a process that contrasts with humans' natural writing and reasoning process, which is often non-linear and hierarchical. While this mismatch is well-documented, the working assumption has been that architectural changes are needed to address it. We argue that by simply rearranging and modifying the training data, autoregressive modeling can more accurately imitate some aspects of the true data-generating process without any changes to the architecture or training infrastructure. We introduce TRELAWNEY, a purely data-centric method that modifies the training data by interleaving sequences with special lookahead tokens that contain future information. This simple data augmentation, requiring no changes to model architecture or training infrastructure, equips models to both condition on future goals and generate them. We present representative results on high-entropy tasks like path planning, algorithmic reasoning, zebra puzzles, and controllable generation, demonstrating improved performance on tasks with branching paths or long-horizon planning. Finally, our method enables the generation of plausible long-term goals at no additional cost, potentially opening doors to new capabilities beyond the current language modeling paradigm.

## 1 INTRODUCTION

Next-token prediction (NTP) is the primary objective for training sequence models. This objective involves a technique called *teacher forcing* (Williams & Zipser, 1989), where the model's predicted output at each step is replaced with the ground truth from the real dataset. One of teacher forcing's benefits is that it accelerates training by providing the model with the correct previous output, so the learning does not suffer from error accumulation, and the gradient update is more stable. Another crucial benefit is that it enables parallelism and hardware acceleration in training because the model can simultaneously process all time steps, rather than sequentially waiting for its own predictions. However, Bachmann & Nagarajan (2024) argue that models trained with teacher forcing often fail to learn long-range dependencies, latching onto local patterns and surface-level correlations instead.

Several recent methods have been proposed to alleviate the issue of teacher forcing. One popular approach is *multi-token prediction*, where the model learns to predict multiple tokens at the same time (Bachmann & Nagarajan, 2024; Gloeckle et al., 2024; Deepseek et al., 2024). Another family of methods involves modifying the training objective to predict both the next token for a prefix and the previous token for a suffix by modifying the model architecture (Hu et al., 2025). Most of these approaches either involve nontrivial modifications or make the learning process much harder by forcing the model to predict multiple tokens at the same time.

In this work, we investigate a data-centric approach to address these limitations. Instead of modifying the model architecture, our method TRELAWNEY[1] modifies the training data by introducing alternative factorizations that embed inductive biases directly. Concretely, we augment the training corpus by interleaving it with special lookahead tokens — `<T>` and `</T>` — that encapsulate future information (see Figure 2).

Our simple model agnostic data-rearrangement procedure results in both improved task performance in domains otherwise difficult for models trained with next token prediction, by decoupling the training objective from the underlying data-generating function the model needs to learn.

---

[1]The name is inspired by the seer who predicts the future in the Harry Potter series.

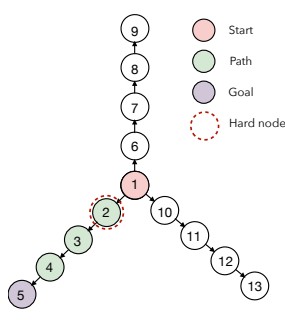 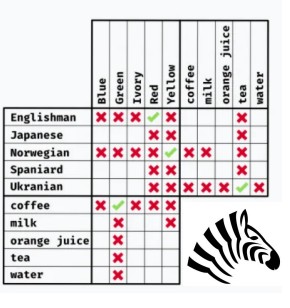

Path Planning · Zebra Puzzles · Story generation

Figure 1: The need for knowledge about the future are inherent to most tasks. In this work, we explore well-known benchmarks that cleanly demonstrate the need for conditioning on future states, and show how TRELAWNEY will enable improvement gains even if only used during training.

## 2 RELATED WORK

**Pitfalls of Next token prediction.** Bachmann & Nagarajan (2024) characterizes two failures that occur in next-token prediction, those that emerge from (1) teacher-forced training, and (2) those emerging at inference, where errors compound. Several prior works (Arora et al., 2022; Ross et al., 2011) have focused on inference-time errors. Our work is more related to the training time failure. During training, the maximum likelihood estimation (MLE) objective treats all tokens equally. However, Bigelow et al. (2024) provides empirical evidence that tokens contribute unequally to overall performance, suggesting that some tokens are inherently more critical than others. Relatedly, Lin et al. (2024) proposes leveraging a stronger model to identify and prioritize these important tokens for more efficient pretraining. Nye et al. (2021) introduces scratchpads to augment the model's input with intermediate reasoning steps for multi-step problem solving. Quiet-STaR Zelikman et al. 2024 similarly encourages models to "think quietly" by sampling token-level internal rationales that explain future text and then optimizing their usefulness for next-token prediction with REINFORCE-style reinforcement learning objectives. Goyal et al. (2023) introduces pause tokens at training and inference, as a mechanism for delayed next-token prediction, which improves performance on language tasks.

Of particular importance is awareness of three pitfalls from Bachmann & Nagarajan (2024): (1) *Clever Hans Cheat.* When training with teacher-forcing, the model is provided with ground truth prefixes (e.g., $v_{\text{start}}, v_1, \ldots, v_{i-1}$) that include parts of the answer. This extra information can enable the model to "cheat" by simply copying the easy tokens that follow without learning the true underlying plan. (2) *Indecipherable Token Problem.* Because the later tokens can be easily predicted using the Clever Hans cheat, the crucial early decision receives insufficient gradient signals. This early token becomes "indecipherable" since its correct prediction relies on long-range planning that is effectively bypassed during teacher-forced training. (3) *Exposure bias.* During inference, the model is likely to make a mistake because the model has not learned the indecipherable token – the model was never trained to rely on its own predictions. The mismatch between training (where the model always sees the correct previous tokens) and inference (where it must rely on its own predictions) can lead to a cascading sequence of errors.

**Non-causal sequence modeling.** offers an alternative to the traditional autoregressive, left-to-right generation constraint by allowing the model to use both past and future context (Gu et al., 2017; Gong et al., 2022; Nolte et al., 2024). Bavarian et al. (2022) propose a "fill in the middle" strategy which changes the data ordering, while T5 (Raffel et al., 2020) incorporates span corruption, $\sigma$-GPT (Pannatier et al., 2024) uses on-the-fly order modulation, MLM-$\mathcal{U}$ (Kitouni et al., 2024) uses uniform masking similar to the diffusion objective and XLNet (Yang et al., 2019) leverages permutation-based training. Inference-time strategies, such as tree generation (Welleck et al., 2019), have also been explored. Beyond language modeling, video prediction (Han et al., 2019; Vondrick et al., 2016) similarly relies on non-causal prediction of future frames or states. In control tasks and world modeling (LeCun, 2022; Hafner et al., 2023; Lin et al.), non-causal approaches provide a more comprehensive representation of environmental dynamics, thereby enhancing long-term planning.

**Controllable generation.** Our work is also related to controllable generation, where the models are conditioned to follow goals or guidelines provided through explicit instructions or auxiliary inputs. Prominent methods include Keskar et al. (2019); Dathathri et al. (2019); Krause et al. (2020), and

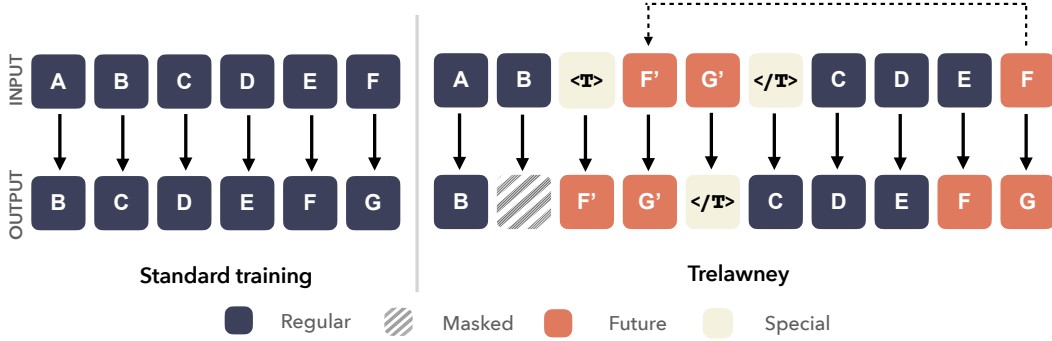

Figure 2: TRELAWNEY. Inserting tokens from the future helps the model capture otherwise diffuse long-distance relationships. The future, delimited with special tokens `<T>` and `</T>`, is incorporated into the modified sequences, so the model is encouraged to learn what it will generate in the future (i.e., `F' G'`) and the path leading there (i.e., `CDE`), making the actual future (i.e., `FG`) easier to predict.

*prompting* (Brown et al., 2020; Wei et al., 2022). In comparison, TRELAWNEY does not require a curated dataset or additional classifiers and achieves fine-grained temporal control.

## 3   TRELAWNEY

Consider a sequence of tokens $\mathbf{y} = (y_1, y_2, \ldots, y_T)$, where each token $y_t$ belongs to a fixed vocabulary $V$, and that $\mathbf{y}$ follows a distribution $P(\mathbf{y})$. An auto-regressive model $P_\theta$ factorizes the joint probability of $\mathbf{y} = (y_1, y_2...y_T)$ as: $P_\theta(\mathbf{y}) = \prod_{t=1}^{T} p_\theta(y_t \mid \mathbf{y}_{<t})$ where $\mathbf{y}_{<t} = (y_1, \ldots, y_{t-1})$ denotes all tokens before index $t$. In next token prediction, we maximize the likelihood of each token under the ground truth context (teacher forcing) from the training corpus. At inference, the model predicts the next token by sampling or selecting the most likely token, conditioned on an optional context $\mathbf{c}$ (e.g., question). In the greedy setting, the next token $\widehat{y}_t$ is $\hat{y}_t = \arg\max_{y_t} p_\theta(y_t \mid \widehat{\mathbf{y}}_{<t}, \mathbf{c})$, where, $\widehat{\mathbf{y}}_{<t}$ denotes the model's own generated tokens.

TRELAWNEY is a data augmentation scheme that modifies the given sequence $\mathbf{y}$ as follows: first select a point $d$ and insert a sequence of $k$ *conditioning tokens*, $\mathbf{z} = (z_1, z_2, \ldots, z_k)$, delimited with special tokens `<T>` and `</T>`. Concretely, we have the following augmentation:

$$(y_1 \ y_2 \ \ldots \ y_T) \Longrightarrow (y_1 \ y_2 \ \ldots \ y_d \ \texttt{<T>} \ \mathbf{z} \ \texttt{</T>} \ y_{d+1} \ \ldots \ y_{T-1} \ y_T).$$

The choices of $d$, $k$, and the content of $\mathbf{z}$ are flexible, and we present several strategies that can alleviate the problems of existing models. This provides an easy mechanism for model designers and practitioners to inject domain knowledge. However, domain knowledge is *not necessary* to see the benefits of TRELAWNEY. Our experiments will show that even *randomly chosen* conditioning tokens from the future are often sufficient to resolve the aforementioned issues of NTP.

### 3.1   ENCODING DECISION POINTS & FUTURES

Our strategies for choosing $\mathbf{z}$ are as follows:

**Copying.** We can directly copy a part of the sequence from a point after $y_d$ to between the special tokens. For $s$ such that $d < s \leq T - k$, the conditioning tokens are the subsequence $\mathbf{y}_{s:s+k}$, resulting in

$$\tilde{\mathbf{y}}_{\text{copy}} \equiv y_1 \ y_2 \ \ldots y_d \ \texttt{<T>} \ \mathbf{y}_{s:s+k} \ \texttt{</T>} \ y_{d+1} \ \ldots \ y_{T-1} \ y_T.$$

The choice of conditioning tokens can have a significant impact on the behavior of the resulting model. For certain types of data, there are *decision points* where there are many different possible futures. These points are good candidates for choosing $d$. At such points of high uncertainty, conditioning on specific possible futures allows for more controllable long horizon planning. Analogously, we can choose the conditioning tokens to be *future tokens* that indicate which future is being generated. Our experiments will outline ways $z$ might be chosen (§4.1, §4.2, §4.3), but even random selection of $d$, $s$, and $k$ will yield benefits.

**Positional information.** In the previous approach, $d$ and $s$ can vary between different data points. This can be problematic if two sequences have very different values of $s - d$. Intuitively, this makes the modeling task harder because there may be conflicting information between different sequences. For example, suppose $\mathbf{y}^1$ and $\mathbf{y}^2$ share the same prefixes, $\mathbf{y}^1_{:d} = \mathbf{y}^2_{:d}$ but the relevant future tokens

are at locations with large differences. To mitigate this conflict, we introduce additional *positional information* into the future tokens, $\zeta(k, \mathbf{z})$. For example, we can have:

$$\zeta(k, \mathbf{z}) = \text{``I want the [k]th sentence from here to be } \mathbf{z}\text{''},$$

$$\tilde{\mathbf{y}}_{\text{copy+pos}} \equiv y_1 \, y_2 \, \ldots \, y_d \texttt{ <T> } \zeta(k, \mathbf{z}) \texttt{ </T> } y_{d+1} \, \ldots \, y_{d+k} \, \ldots \, y_n.$$

Once again, the exact design of the positional information can be problem-dependent (§ 4.4), but does not need to be highly accurate as long as it reduces potential conflict. Similarly, the copied text $\mathbf{z}$ can be a copy of a sequence from the future, $\mathbf{y}_{d:d+k}$, but does not need to be identical, so long as it contains relevant information (e.g., paraphrase). We express $\zeta$ in natural language because this allows the model to integrate $\zeta$ with its pretraining knowledge and also lets the user specify different goals.

### 3.2 DATASET CONSTRUCTION AND TRAINING OBJECTIVE

**Dataset construction.** We want to introduce additional capabilities via the augmentation schema without hurting the traditional language modeling ability of the model. To accomplish this, we train on both regular text and augmented text simultaneously. Specifically, given an original dataset $D = \{\mathbf{y}^{(i)}\}_{i=1}^N$ and an augmentation schema $\texttt{aug}$. We can construct a distribution for the original dataset and a distribution for the augmented dataset:

$$\mathcal{D}(\mathbf{s}) = \frac{1}{N} \sum_{i=1}^N \mathbb{I}\left\{\mathbf{s} = \mathbf{y}^{(i)}\right\}, \quad \mathcal{D}_{\text{aug}}(\mathbf{s}) = \frac{1}{N} \sum_{i=1}^N \mathbb{I}\left\{\mathbf{s} = \texttt{aug}(\mathbf{y}^{(i)})\right\}.$$

For a probability $p$ that controls how much of the training distribution comprises the original data, the training distribution is the following mixture: $\mathcal{D}'(\mathbf{s}) = p\,\mathcal{D}(\mathbf{s}) + (1-p)\,\mathcal{D}_{\text{aug}}(\mathbf{s})$.

**Training and loss function.** During training, the model parameters are optimized using a standard cross-entropy loss with teacher forcing on $\mathcal{D}'$. This allows us to take advantage of all existing engineering optimizations for training language models. One caveat for training with the new dataset $\mathcal{D}'$ is that choosing the decision point and future tokens arbitrarily will result in a large portion of sequences with the next token being $\texttt{</T>}$ at arbitrary locations.

This would distract from the learning process and does not help learning the underlying distribution, since the special tokens are synthetically introduced. Instead, we modify the regular cross-entropy loss by masking the special start token, $\texttt{<T>}$:

$$\mathcal{L}(\mathcal{D}') = -\mathbb{E}_{\mathbf{y}\sim\mathcal{D}'}\left[\frac{1}{|\mathbf{y}|}\sum_{j=1}^{|\mathbf{y}|}\mathbb{I}\{y_j \neq \texttt{<T>}\}\log P(y_j \mid \mathbf{y}_{<j})\right].$$

Here, $\mathbb{I}\{y_j \neq \texttt{<T>}\}$ ensures no loss is computed for the prediction of the special token $\texttt{<T>}$. Note that we do not exclude the loss on $\texttt{</T>}$ because there is a utility to predicting the closing of the future tokens, which we will elaborate on below.

### 3.3 INFERENCE

With TRELAWNEY, we can have two distinct modes of generation.

**Standard autoregressive generation.** The model generates sequences autoregressively without any intervention, following any standard decoding algorithm.

**<T>-generation.** We aim to enable the model to explicitly consider future context at appropriate decision points, to improve its ability to plan ahead. At each decision point $y_d$ in sequence generation, we explicitly insert the special token $\texttt{<T>}$. Subsequently, **(a)** either the model generates the sequence $\mathbf{z}$ autonomously, enabling it to create plausible future plans, or **(b)** incorporates a user-specified sequence $\mathbf{z}$, enhancing controllability. Recall that during the training process, we compute the loss on the $\texttt{</T>}$ token; this allows the model to generate future goals, which can then be used for conditional generation. In contrast to existing methods such as Hu et al. (2025) that require specific decoding mechanisms, our approach can use any off-the-shelf decoding algorithm.

## 4 EXPERIMENTS

We evaluate the effectiveness of TRELAWNEY in both fine-tuning and pretraining settings. Our primary goal is to assess whether augmenting data with lookahead tokens improves a model's capacity for long-range planning, reasoning, and controllable generation.

**Finetuning:** We use four targeted benchmarks designed to isolate specific challenges. We begin with synthetic tasks that offer a controlled environment for analysis and then proceed to a natural language task to test for broader applicability.

- **Star Graph (§ 4.1):** A task designed to highlight a known failure mode of standard next-token prediction in simple, long-range dependency settings.
- **Algorithmic Reasoning (§ 4.2):** A benchmark that requires multi-step, structured reasoning. This tests whether future anchor points improve the model's ability to follow complex procedures.
- **Zebra Puzzles (§ 4.3):** A constraint-satisfaction problem with long-range, cross-coupled clues that require models to have global consistency and low cascading errors.
- **Story Generation (§ 4.4):** A creative generation task that requires high-level planning and fine-grained, user-directed control over the narrative structure.

These fine-tuning experiments are designed to answer the following questions:

- **RQ1:** Does training with TRELAWNEY improve performance on the downstream task during standard autoregressive inference (i.e., without any lookahead tokens provided)?
- **RQ2:** Does providing a ground-truth or user-specified lookahead sequence z at inference time improve task performance and grant users explicit control over the generation?

**Pretraining:** To evaluate the generalizability and broader utility of our approach, we apply TRELAWNEY during the pretraining of a language model on a large-scale corpus. This setting allows us to investigate whether the benefits observed in fine-tuning transfer to a general-purpose foundation model. Specifically, we seek to answer:

- **RQ3:** How does pretraining with TRELAWNEY impact performance on standard language modeling benchmarks and downstream tasks when using standard autoregressive generation?
- **RQ4:** After pretraining, does the model retain a general ability to perform lookahead-conditioned generation on novel, unseen prompts and tasks?

### 4.1 PATH PLANNING

The star graph is a simple path-finding problem introduced by Bachmann & Nagarajan (2024), where, given a directed graph $G(d, n)$ with degree $d$ and path length $n$, the objective is to find a path from the start node to the goal node (Figure 3). Despite its simplicity, traditional next-token prediction (NTP) struggles on this task. A key challenge is that the critical decision point occurs at $v_1$, the first node after $v_{\text{start}}$. This node is hard to predict because $v_{\text{start}}$ has many outgoing edges. As discussed in Section 2, teacher forcing can lead to undesirable behavior on this simple dataset.

**Dataset and Augmentation Schema.** To mitigate these issues, we introduce a future subgoal $\mathbf{z}$, as any contiguous subsection of the path in $[v_2, v_{\text{goal}})$. This modification compels the model to generate a meaningful intermediate plan rather than simply copying the full ground truth prefix. As a result, the model receives a stronger learning signal for critical early decision-making. Each example $\mathbf{y} = (\mathbf{p}, \mathbf{c})$ in the dataset is a prefix and completion pair. The prefix $\mathbf{p}$ is given by the adjacency list of G followed by the $v_{start}, v_{goal} =$. The completion $\mathbf{c}$ is the path $v_{\text{start}}, v_1, v_2, ... v_{\text{goal}}$, i.e., $\mathbf{p} \equiv \text{Adj}(G) \mid v_{\text{start}}, v_{\text{goal}} =$ and $\mathbf{c} \equiv v_{\text{start}}, v_1, v_2, \ldots, v_{\text{goal}}$.

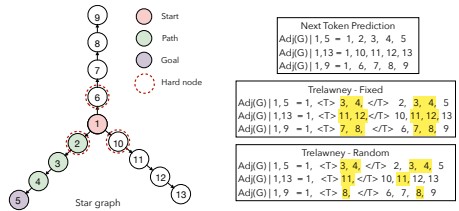

Figure 3: In the star graph, there are key "hard nodes" that indicate the moment of branching, after which the path and goal become clear. Above is a visualization of the construction and linearizations of $\mathcal{D}'$ for the star graph.

Our task augmentation schema $\mathbf{y} \implies \tilde{\mathbf{y}}_{\text{copy}}$ is:

$$\mathbf{y} \equiv \text{Adj}(G) \mid v_{\text{start}}, v_{\text{goal}} = v_{\text{start}}, v_1, v_2, \ldots, v_{\text{goal}},$$
$$\tilde{\mathbf{y}}_{\text{copy}} \equiv \text{Adj}(G) \mid v_{\text{start}}, v_{\text{goal}} = v_{\text{start}}, \texttt{<T>} \, \mathbf{z}, \texttt{</T>} \, v_1, \ldots, v_{\text{goal}}.$$

**Choice of z.** We vary $\mathbf{z}$ (a contiguous subsequence of future tokens) across experiments and ablations (see Figure 3). Its role is to guide planning by indicating a subgoal on the path from $v_{\text{start}}$ to $v_{\text{goal}}$. We exclude $v_1$ to avoid the Clever Hans cheat discussed above. We also exclude $v_{\text{goal}}$ so that the model learns the long-term dependency between start and goal without having direct access to the goal token. An ablation study confirms that including $v_{\text{goal}}$ does not yield further improvements.

| | Path planning G(*,*) | | | | Alg Reasoning scc- | | | | | |
| --- | --- | --- | --- | --- | --- | --- | --- | --- | --- | --- |
| | G(2,5) | G(5,5) | G(20,5) | G(2,10) | | scc-4 | scc-5 | scc-11 | scc-12 | scc-15 |
| **NTP** (AutoReg.) | 0.50 | 0.20 | 0.05 | 0.50 | | 1.00 | 0.99 | 0.62 | 0.57 | 0.27 |
| **TRELAWNEY** | | | | | | | | | | |
| – Fixed | 1.00 | 1.00 | 1.00 | 0.52 | – Rule-Based | 1.00 | 1.00 | 0.73 | 0.62 | 0.31 |
| – Random | 1.00 | 1.00 | 1.00 | 0.50 | – Random | 1.00 | 0.978 | 0.718 | 0.706 | 0.476 |
| – Fixed (Gen.) | 1.00 | 1.00 | 1.00 | 0.57 | – Rule-Based | 1.00 | 1.00 | 0.73 | 0.65 | 0.34 |
| – Random | 1.00 | 1.00 | 1.00 | 0.91 | – Random | 1.00 | 0.998 | 0.776 | 0.79 | 0.512 |
| – Fixed (Spec.) | 1.00 | 1.00 | 1.00 | 1.00 | – Rule-Based | 1.00 | 1.00 | 0.84 | 0.76 | 0.47 |
| – Random | 1.00 | 1.00 | 1.00 | 0.91 | – Random | 1.00 | 0.998 | 0.828 | 0.812 | 0.544 |

Table 1: TRELAWNEY shows strong performance in both Path Planning and Algorithmic Reasoning across all problem complexities compared to NTP. This result is consistent across all 3 modes of generation: simple autoregression (AutoReg), a model Generated goal (Gen), or a user Specified goal (Spec). Notably, random augmentation often outperforms fixed or rule-based augmentation.

**Training.** Data is generated programmatically via the official implementation. Although we use pretrained models, each node remains a single token in the tokenizer. Models are trained on $200,000$ examples (§ A.3) with standard teacher forcing training and two augmentation schemas.

TRELAWNEY-fixed: In a single training run, the choice of $\mathbf{z}$ is fixed across all examples. Specifically, $\mathbf{z}$ is chosen as a contiguous sequence of 1 to 4 nodes with a fixed start and end point across all sequences in the dataset (Figure 3).

TRELAWNEY-random: $\mathbf{z}$ can vary between examples. We randomly select any contiguous subsequence of the path after $v_1$ to serve as $\mathbf{z}$ in $\tilde{\mathbf{y}}_{\text{copy}}$. We do not include $v_1$ (the hard node) as part of $\mathbf{z}$ (Figure 3). Without fixed positional information, the model learns to generate its own goals of varying lengths. We observe that this variant is successful in solving longer planning problems.

**Evaluation (§A.2.1).** We evaluate the models on 5,000 held-out examples for each graph, reporting the accuracy of the generated path compared to the ground truth.

**Results.** In Table 1, we see that all variants of TRELAWNEY outperform next token prediction. Additionally, on shorter graphs $G(2,5)$, $G(5,5)$, $G(10,5)$, $G(20,5)$, training with TRELAWNEY improves autoregressive generation at no additional cost, suggesting that the model implicitly learns to plan better (possibly due to pre-caching or breadcrumbs proposed by Wu et al.) and can generate long-term goals. For longer graphs $G(2,10)$, the TRELAWNEY-random variant can complete the task when the model is used to generate its own subgoal sequence $\mathbf{z}$, indicating that model-generated goals can improve planning and do not require specialized knowledge for choosing $\mathbf{z}$.

TRELAWNEY-random is notably more performant on graphs with longer paths when compared to TRELAWNEY-fixed. Both variants of TRELAWNEY succeed when user-provided goal sequences are provided, showing that explicit goal hints allow for better controllability. Further, ablations conducted on larger models (See A.5) show that planning abilities improves with model capacity.

### 4.2 ALGORITHMIC REASONING

CLRS-Text (Markeeva et al., 2024) is a benchmark of algorithmic reasoning. The input is the algorithm name, followed by a step-by-step reasoning trace and the final answer. We pick a representative example from algorithms that require backtracking, i.e., tasks that benefit from information of future states. We choose strongly-connected-components, a step-by-step sequential prediction task where each step is longer than one token, and report results on it. The trace contains the execution of Tarjan's algorithm (Tarjan, 1972), which computes strongly connected components in linear time by performing a depth-first search that tracks low-link values and uses a stack to detect cycles.

**Dataset and Augmentation Schema.** In each example $\mathbf{y} = (\mathbf{p}, \mathbf{c})$ of the strongly-connected-components subset, the prefix $\mathbf{p}$ is given by the adjacency matrix of the initial graph. The completion $\mathbf{c}$ is graph execution traces of the algorithm followed by the final answer, i.e., $\mathbf{p} \equiv \text{Adj}(G) =$ and $\mathbf{c} \equiv t_1, t_2 \ldots t_n | F$ where $t_i$ is the state of the graphical trace and $F$ is the final answer. Our augmentation schema $\mathbf{y} \implies \tilde{\mathbf{y}}_{\text{copy}}$ for this task is as follows:

$$\mathbf{y} \equiv \text{algo: } \text{Adj}(G) = t_1, t_2, \ldots, t_n | F,$$
$$\tilde{\mathbf{y}}_{\text{copy}} \equiv \text{algo: } \text{Adj}(G) = t_1, \texttt{<T>} \, \mathbf{z} \, \texttt{</T>} \, t_2, \ldots, t_n | F.$$

Unlike the star graph task — where failure typically occurs at a single critical decision point — the algorithmic reasoning tasks involve multiple branching points where errors can accumulate. In the strongly connected components subset, the state sequence $t$ represents the graph execution trace and comprises multiple tokens, each corresponding to a distinct graph state. By segmenting the trace into these meaningful units, our augmentation schema is better able to capture intermediate reasoning steps and guide the model's planning process throughout the entire execution trace.

**Choice of z.** For simplicity, we fix the decision point $y_d$ at the second state in each trace. In algorithmic reasoning tasks do not present a clear failure point — there can be many points in the trace at which misprediction causes the entire generation to diverge. We only pick $\mathbf{z}$ as a complete step $t_i$ in the trace and how $i$ is determined for each variant. Gains here further demonstrate generality as domain-specific knowledge is not required to see performance improvement.

**Training.** Data for all experiments are sub-selected from the original dataset. We train a single model on problems of varying sizes. Since we do not test for length generalization, we only report accuracies on sizes present in the training corpus (60K samples). We train two variants (§C.1):

TRELAWNEY-rule-based: For every example in $\mathcal{D}_{\text{aug}}$, $\mathbf{z}$ is chosen as the first change in the trace provided. The position of $\mathbf{z}$ in the trace varies across graph sizes and graphs.

TRELAWNEY-random: $\mathbf{z}$ is chosen as a single random state in the trace provided. per graph length. We evaluate the models similar to the star-graph setting, and report the accuracies of the final answer.

**Evaluation.** We evaluate on 500 examples (CLRS-Text-test)

**Results** In addition to strong results as task complexity increases, in App. Figure 7 we show a trend that TRELAWNEY-random consistently improves on next token prediction when using `<T>`-generation and, surprisingly, in standard autoregressive generation as well. TRELAWNEY-rule-based although being chosen more strategically, performs worse than `<T>`-random.

### 4.3 ZEBRA PUZZLES

Zebra (Einstein) puzzles are a constraint satisfaction problem specified by clues over $m$ entities (e.g., houses) and $n$ attributes (e.g., color, nationality, pet), (Figure 1); the goal is to deduce a unique grid assignment that satisfies all clues. We follow the symbolic formulation given by Shah et al. (2024).

**Dataset and Augmentation Schema** Each example $\mathbf{y} = (\mathbf{p}, \mathbf{c})$ pairs a clue set $\mathbf{p}$ with a completion trace $\mathbf{c}$. Here, $\mathbf{p}$ is the puzzle's set of clues defining the constraints, and $\mathbf{c} \equiv t_1, \ldots, t_T \mid F$, where each step $t_i = (r_i, c_i, v_i)$ assigns value $v_i$ to row $r_i$ and column $c_i$; $F$ is the final solution grid satisfying all clues. Our augmentation $\mathbf{y} \Longrightarrow \tilde{\mathbf{y}}_{\text{copy}}$ for this task is similar to algorithmic reasoning :

$$\mathbf{y} \equiv \mathbf{p} = t_1, t_2, \ldots, t_n | F,$$
$$\tilde{\mathbf{y}}_{\text{copy}} \equiv \mathbf{p} = t_1, \texttt{<T>} \, \mathbf{z} \, \texttt{</T>} \, t_2, \ldots, t_n | F.$$

A key difference is that this problem in NP-hard and the choice of which constraints to enforce first can make the problem easier or harder to solve. In our experiments we use a solver generated ordering for the trace, which is the simpler choice of steps.

**Choice of z.** As with 4.2, we fix the decision point $y_d$ at the second state in each trace. We pick $\mathbf{z}$ as a set of continuous complete steps $t_i..t_i + n$ in the trace. We randomize this choice to further test generality, without domain specific knowledge.

**Training.** We train on 1,000,000 examples with class balanced samples of 62,500 for each of the 16 different classes of problems.

**Evaluation.** We evaluate on 800 examples with 50 examples each of puzzles of 16 complexities. We report the accuracy on the entire puzzle.

**Results.** Autoregressive generation with TRELAWNEY-random (35.75%) outperforms next token prediction (30.1%). With user specified goals, TRELAWNEY improves the final accuracy to 53.5% (See Fig. 10). Interestingly, model generated goals are often hard to predict and show lower accuracy (9.75%), indicating that for problems with large branching factor, with few high entropy points can still be hard to model with only random slices of lookaheads. Remarkably, models trained with TRELAWNEY solve problems of higher complexity (See Tab. 6, Fig. 9) , which typically also have longer traces and may point towards better long horizon planning.

**PROMPT** Once upon a time, there was a little girl named Lily. Lily loved to play in the rain.

FUTURE

<T> I want the **[2]**-nd sentence from here to be: **"Hello little frog!** </T>

<T> I want the **[4]**-th sentence from here to be: **"Hello little frog!** </T>

<T> I want the **[8]**-th sentence from here to be: **"Hello little frog!** </T>

GENERATION

One day, Lily was playing in the rain and she saw a little frog. **"Hello little frog!** What are you doing here?" Lily asked. The frog replied, "I am looking for a friend to play with. Can you be my friend?" Lily was happy to have a new friend. She said, "Yes, I can be your friend. We can play in the rain together." So, the frog and Lily played in the rain, making puddles and having fun. At the end of the day, they said goodbye and promised to play again the next day.

One day, Lily went outside to play. She saw a big frog on a log. The frog looked sad. **"Hello little frog!** Why are you sad?" Lily asked. The frog replied, "I can't jump high. I need help." Lily thought for a moment and said, "I have an idea! Let's play together and I will help you jump high." The frog was so happy and thanked Lily. They jumped high and played all day. From that day on, Lily and the frog became best friends.

One day, Lily went outside to play in the rain. She jumped in the puddles and laughed. She loved the feeling of the rain on her face. Suddenly, Lily saw a frog in a puddle. The frog was small and wet. Lily wanted to help the frog. She picked up the frog and put it on a rock. **"Hello little frog!** You are so wet and cold," said Lily. The frog smiled and thanked Lily for helping. From that day on, Lily and the frog became good friends.

Figure 4: Illustration of TRELAWNEY's effect during generation. The top is the prompt, and the middle is different future tokens. The generations are coherent and read naturally.

## 4.4 NATURAL LANGUAGE PLANNING

Finally, we explore the open-ended language task of story generation. Here, random futures are probably not of interest as the user will specify their target, but the model should still be required to sample feasible conclusions. These properties make story generation inherently a planning task, as it requires the model to set long-term goals, maintain consistency, and control narrative flow. Although many narratives include complex elements like detailed backstories and conflict resolution, even simpler story generation requires planning to produce coherent and logically structured output. We evaluate our approach on the **Tiny Stories** dataset from Eldan & Li (2023). In this task, the aim is to generate coherent stories conditioned on specified goals (Fig. 4). This benchmark tests whether our strategy of inserting future tokens can enhance planning in natural language generation.

**Dataset and Augmentation Schema.** Each example $\mathbf{y} = (\mathbf{p}, \mathbf{c})$ is a prefix–completion pair extracted from a story. We use a sentence parser to segment each story into individual sentences or phrases. If a story is split into sentences $s_1, s_2, \ldots, s_n$, the prefix $\mathbf{p}$ is the beginning of the story (e.g., $s_1 \ s_2$) and the completion $\mathbf{c}$ is the remainder (i.e., $s_3 \ s_4 \ \ldots \ s_n$).

Our augmentation schema $\mathbf{y} \implies \tilde{\mathbf{y}}_{\text{copy+pos}}$ is defined as:

$$\mathbf{y} \equiv s_1 \ s_2 \ \ldots \ s_n,$$
$$\tilde{\mathbf{y}}_{\text{copy+pos}} \equiv s_1 \ s_2 \ \ldots \ s_d \text{ <T> } \zeta(k, s_{d+k}) \text{ </T> } s_{d+1} \ \ldots \ s_{d+k} \ \ldots \ s_n,$$
$$\zeta(k, s) = \text{"I want the [k]-th sentence from here to be [s] ".}$$

**Choice of** $\zeta(k, s)$**.** We choose decision points randomly at the end of the $k$-th sentence in the document, as the position to to insert $\zeta(k, s)$. The subgoal [s] is defined in $\zeta(d, s)$ as extracted from the corresponding sentence $s_{d+k}$.

**Training.** All models are trained on $300,000$ examples from the Tiny Stories dataset for 1 epoch using the masked cross-entropy loss specified in § 3.2 (See App. A.3).

**Evaluation**: We follow the evaluation protocol used by Hu et al. (2025) and use GPT-4 to rate 100 generated stories from each model. The stories are anonymized and shuffled to prevent any information leakage about the author. Each evaluation is repeated over 6 trials. We report the win rate with binomial confidence intervals computed at a 95% significance level. (See Fig. 5, E.3)

**RQ1: Does TRELAWNEY improve goal reaching ability i.e., resulting in more controllable generation?** We compare the completions from few-shot prompts on the baseline with those obtained by explicitly specifying goals on TRELAWNEY-implicit. Qualitatively, we observe that models trained with TRELAWNEY generate stories that more effectively reach the intended long-term goals (see Figure 2). Quantitatively, GPT-4 prefers TRELAWNEY to few-shot prompts on next-token-prediction, 76.53% of the time, with a confidence interval of [72.9%, 79.9%]. This result suggests that TRELAWNEY is much

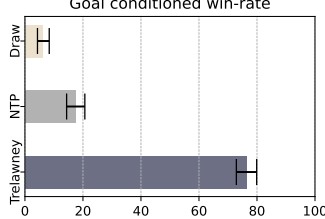

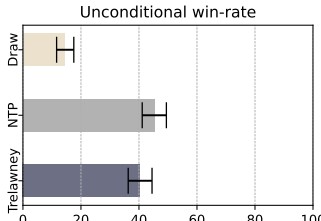

Figure 5: Our story generation evaluation demonstrates greatly improved performance when goal-conditioned, without hurting unconditional generation.

| | ArcEasy | HellaSwag | Lambada | LogiQA2 | MMLU | PIQA | SciQ | WinoGrande | Avg |
|---|---|---|---|---|---|---|---|---|---|
| NTP-124M | 0.467 | 0.302 | 0.335 | 0.216 | 0.229 | 0.669 | 0.732 | 0.510 | 0.433 |
| TRELAWNEY-124M | 0.449 | 0.308 | 0.348 | 0.227 | 0.229 | 0.669 | 0.724 | 0.506 | 0.433 |
| NTP-1B | 0.572 | 0.413 | 0.509 | 0.220 | 0.234 | 0.728 | 0.837 | 0.549 | 0.508 |
| TRELAWNEY-1B | 0.578 | 0.415 | 0.505 | 0.220 | 0.249 | 0.729 | 0.830 | 0.549 | 0.509 |

Table 2: No performance decrease on zero-shot evaluation across standard downstream evaluations.

more effective at controllable generation than few-shot and natural language prompting. We provide more details and ablations of prompting variants used in E.2.

**RQ2: Preference on stories generated by standard autoregressive generation.** We compare the standard autoregressive generations from models trained with TRELAWNEY and models trained with NTP. Quantitatively, we observe that GPT-4 prefers autoregressive generations on TRELAWNEY to next-token-prediction, $40.35\%$ of the time, with a binomial confidence interval of [44.5%, 36.2%]. The justification for judgements appear to be preferences in ending of the stories, which qualitatively, does not appear to affect factors such as coherence and creativity. We provide examples of GPT-4 preference evaluations in E.2. We evaluate perplexity on Wikitext (Merity et al., 2016) to verify that TRELAWNEY maintains language model performance, with results comparable to the baseline (§E.4).

## 4.5 PRETRAINING

Pretraining provides a more general setting to evaluate if TRELAWNEY's benefits extend beyond task-specific finetuning. Unlike synthetic tasks where clear decision points can be identified or constrained natural language tasks, large-scale pretraining lacks obvious choice points. This makes it a challenging, but important test of whether the augmentation can improve long-horizon planning without degrading standard language modeling performance.

**Dataset and Augmentation Schema:** We pretrain decoder-only transformers on 10B tokens of Fineweb (Penedo et al., 2024). Following the schema from § 4.4, we augment documents with probability $p = 0.5$. For each augmented document, an insertion index is selected uniformly at random every 35 sentences, and the future span **z** is chosen as a sentence sampled between 2 and 8 sentences ahead of the insertion point. 124M and 1B parameter models are trained with 10B tokens.

**Evaluation.** We evaluate two aspects of performance. First, we evaluate standard autoregressive generation by comparing zero-shot downstream accuracy across standard language modeling benchmarks - ArcEasy (Clark et al., 2018), HellaSwag (Zellers et al., 2019), Lambada (Paperno et al., 2016), LogiQA2(Liu et al., 2023), MMLU (Hendrycks et al., 2020), PIQA (Bisk et al., 2020), SciQ (Johannes Welbl, 2017) and WinoGrande (Sakaguchi et al., 2021) and by reporting perplexity on datasets from the Paloma (Magnusson et al., 2024) suite. This allows us to test whether the augmentation preserves baseline language modeling ability. Second, we evaluate conditional generation following the same protocol used in §4.4, where explicit goals are provided during inference (F.2)

**Results.** TRELAWNEY maintains language modeling quality, with perplexity comparable to next-token prediction (See Table 9). On downstream tasks, performance is unchanged for 124M models and shows marginal gains for 1B models. Conditional generation confirms that explicit goals can guide long-horizon planning without reducing fluency. These preliminary findings suggest that the augmentation scales naturally to pretraining and may yield larger benefits at a greater scale. Prior work on multi-token prediction (Gloeckle et al., 2024) shows that some methods become more effective as models grow. A more detailed analysis across domains is beyond the scope of this paper.

## 5 DISCUSSION

The machinery of autoregressive language modeling is flexible and highly efficient, but autoregressive modeling is not the most natural choice for many sequence modeling tasks, as we discussed at the beginning. By simply augmenting training with future states (and training appropriately as we outline), models can overcome many of the known challenges of next-token prediction and even be imbued with better controllable generation.

Our finetuning experiments are chosen to directly identify branching and planning complexity and show the effectiveness of even the TRELAWNEY-random augmentation at improving models. Further, our pretraining results indicate that the approach could extend to broader domains and be integrated into standard pipelines without harming performance. Beyond simple copying behaviors, our method opens the door to future research using reinforcement learning to control generation based

on the information enclosed by the special tokens. One remaining challenge is determining when the model should leverage these capabilities; uncertainty metrics may offer a promising solution to identify lookahead points.

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

# A  APPENDIX

## A.1  REPRODUCIBILITY STATEMENT

Code and datasets for all experiments are currently in preparation and will be released.

## A.2  EVALUATION DETAILS

### A.2.1  PATH PLANNING

For the next-token prediction baseline, we evaluate the model using standard autoregressive generation. For models trained with TRELAWNEY, We evaluate both standard autoregressive and `<T>`-generation and compare to a next-token prediction baseline. In the conditional setting, the model uses either model-generated $\mathbf{z}$'s as goals or user-provided ground truth "future goals" as hints. Standard autoregressive generation allows us to test whether TRELAWNEY improves regular generation. `<T>`-generation demonstrates whether the model has learned to generate plausible future goals and use these goals for better planning. By providing intermediate hints, we evaluate if the model can leverage these cues to solve the larger planning problems.

## A.3  IMPLEMENTATION DETAILS

**Training details**: All results are reported on the pretrained-Llama 3.2-1B model. We conducted experiments by sweeping over learning rates of 1e-5, 2e-5, and 1e-6, using the AdamW optimizer with a linear learning rate scheduler for one epoch, and reporting the best result. We use the masked cross-entropy loss specified in § 3.2. We use $p = 0.5$ for all experiments. All experiments were run on 4xA6000 GPUs or 4xL40S GPUs. We will also provide the full list of hyperparameters and release code and datasets used.

## A.4  ABLATIONS - AUTOREGRESSIVE ARCHITECTURES

In this section we also compare against other autoregressive architectures. We use mamba as a representative model class for state space models. We observe that using TRELAWNEY-Random improves on next token prediction on state space architectures as well.

| | | **Path planning** G(*,*) | | | |
| | | G(2,5) | G(5,5) | G(20,5) | G(2,10) |
|---|---|---|---|---|---|
| AR. | NTP | 0.50 | 0.20 | **0.05** | 0.50 |
| | TRELAWNEY | **1.0** | **0.998** | 0.049 | 0.50 |
| Gen. | NTP | – | – | – | – |
| | TRELAWNEY | **1.0** | 0.997 | 0.048 | **0.511** |
| Spec. | NTP | – | – | – | – |
| | TRELAWNEY | **1.0** | **0.998** | 0.048 | 0.50 |

Table 3: Mamba-1.5B - Results on star graph

## A.5  ABLATIONS - MODEL SIZING

To compare the effects of model size on TRELAWNEY-Random, we perform on 0.5B (Qwen2.5-0.5B), 1B (Llama-3.2-1B) and 3B (Llama-3.2-3B) models. We do not account for architectural differences between the Qwen 0.5B model and the 1B and 3B Llama models.

The smallest model is unable to solve the longest graph that we test for $G(2, 10)$, while the 1B model is able to solve the graph when allowed to generate $\mathbf{z}$. Finally, the 3B model, is able to solve the graph with only autoregressive generation when trained with TRELAWNEY. This hints at TRELAWNEY being more effective on larger models, potentially learning better representations, and being easily scalable. Interestingly, larger models can solve the simplest graphs ($G(2, 5), G(5, 5)$) autoregressively. We speculate that this could be due to pre-caching improving with scale as previously observed by Wu et al.

| | | **Path planning** G(*,*) | | | |
|---|---|---|---|---|---|
| | | G(2,5) | G(5,5) | G(20,5) | G(2,10) |
| AR. | NTP | 0.50 | 0.20 | 0.05 | 0.50 |
| | TRELAWNEY | **1.0** | **1.0** | 0.874 | **0.533** |
| Gen. | NTP | – | – | – | – |
| | TRELAWNEY | 1.0 | 1.0 | 0.847 | 0.514 |
| Spec. | NTP | – | – | – | – |
| | TRELAWNEY | **1.0** | **1.0** | **0.931** | 0.523 |

Table 4: Qwen/Qwen2.5-0.5B

| | | **Path planning** G(*,*) | | | |
|---|---|---|---|---|---|
| | | G(2,5) | G(5,5) | G(20,5) | G(2,10) |
| AR. | NTP | **1.0** | **1.0** | 0.05 | 0.50 |
| | TRELAWNEY | **1.0** | **1.0** | **1.0** | **1.0** |
| Gen. | NTP | – | – | – | – |
| | TRELAWNEY | 1.0 | 1.0 | 1.0 | 1.0 |
| Spec. | NTP | – | – | – | – |
| | TRELAWNEY | 1.0 | 1.0 | 1.0 | 1.0 |

Table 5: meta-llama/Llama-3.2-3B

# B  STAR GRAPH

## B.1  RESULTS - LLAMA-3.2-1B

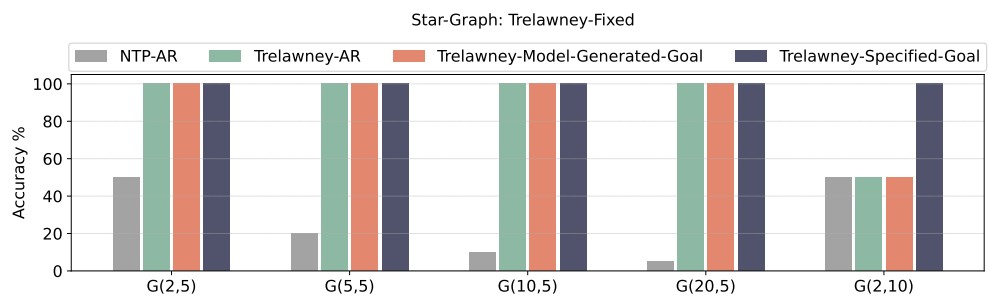

(a) Accuracies TRELAWNEY-Fixed

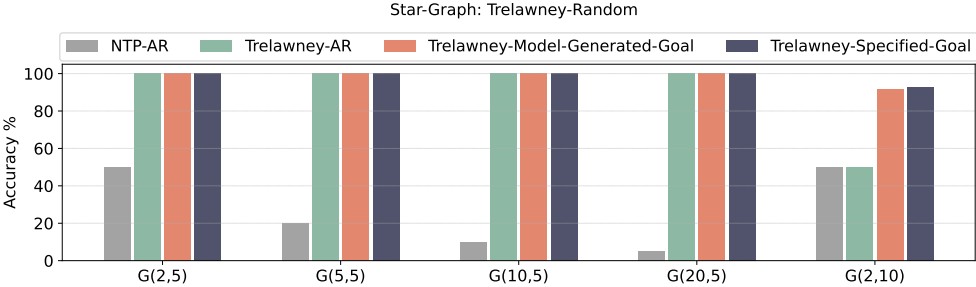

(b) Accuracies TRELAWNEY-Random

Figure 6: Results on Star Graph.

# C ALGORITHMIC REASONING

## C.1 EXAMPLES

We provide examples of the data augmentation schema used in $D_{\text{aug}}$ for the strongly connected components task.Looking at the examples in the data, we see many repeated states $t$ in the trace. The rule used to pick z was likely more beneficial, since the first change in the trace state contains more information than a random trace state.

---

**Strongly connected components - Examples**

**Prefix:**

```
"strongly_connected_components:
 A: [[0 0 0 0 0 0],
     [0 0 0 0 0 0],
     [0 0 0 0 1 0],
     [0 0 0 1 1 0],
     [0 0 1 1 1 0],
     [0 0 0 0 0 1]],
initial_trace: [0 1 2 3 4 5] trace | scc_id:",
```

**Completion:**

```
 "[0 1 2 3 4 5], [0 1 2 3 4 5], [0 1 2 3 4 5], [0 1 2 3 4 5],
  [0 1 2 3 4 5], [0 1 2 3 4 5], [0 1 2 3 4 5], [0 1 2 3 4 5],
  [0 1 2 3 4 5], [0 1 2 3 4 5], [0 1 2 3 4 5], [0 1 2 3 4 5],
  [0 1 2 3 4 5], [0 1 2 3 4 5], [0 1 2 3 4 5], [0 1 2 3 4 5],
  [0 1 2 3 4 5], [0 1 2 3 4 5], [0 1 2 3 4 5], [0 1 2 3 4 5],
  [0 1 2 3 4 5], [0 1 2 3 4 5], [0 1 2 3 4 5], [0 1 2 3 2 5],
  [0 1 2 2 2 5], [0 1 2 2 2 5], [0 1 2 2 2 5], [0 1 2 2 2 5],
  [0 1 2 2 2 5], [0 1 2 2 2 5], [0 1 2 2 2 5], [0 1 2 2 2 5]
  | [0 1 2 2 2 5]"
```

**Completion for TRELAWNEY–Rule-Based:**

```
 "[0 1 2 3 4 5], <T> [0 1 2 3 2 5], </T> [0 1 2 3 4 5],
  [0 1 2 3 4 5], [0 1 2 3 4 5], [0 1 2 3 4 5], [0 1 2 3 4 5],
  [0 1 2 3 4 5], [0 1 2 3 4 5], [0 1 2 3 4 5], [0 1 2 3 4 5],
  [0 1 2 3 4 5], [0 1 2 3 4 5], [0 1 2 3 4 5], [0 1 2 3 4 5],
  [0 1 2 3 4 5], [0 1 2 3 4 5], [0 1 2 3 4 5], [0 1 2 3 4 5],
  [0 1 2 3 4 5], [0 1 2 3 4 5], [0 1 2 3 4 5], [0 1 2 3 4 5],
  [0 1 2 3 4 5], [0 1 2 3 2 5], [0 1 2 2 2 5], [0 1 2 2 2 5],
  [0 1 2 2 2 5], [0 1 2 2 2 5], [0 1 2 2 2 5], [0 1 2 2 2 5],
  [0 1 2 2 2 5], [0 1 2 2 2 5] | [0 1 2 2 2 5]"
```

**Completion for TRELAWNEY–Random:**

```
 "[0 1 2 3 4 5], <T> [0 1 2 3 4 5] </T>  [0 1 2 3 4 5],
  [0 1 2 3 4 5], [0 1 2 3 4 5], [0 1 2 3 4 5], [0 1 2 3 4 5],
  [0 1 2 3 4 5], [0 1 2 3 4 5], [0 1 2 3 4 5], [0 1 2 3 4 5],
  [0 1 2 3 4 5], [0 1 2 3 4 5], [0 1 2 3 4 5], [0 1 2 3 4 5],
  [0 1 2 3 4 5], [0 1 2 3 4 5], [0 1 2 3 4 5], [0 1 2 3 4 5],
  [0 1 2 3 4 5], [0 1 2 3 4 5], [0 1 2 3 4 5], [0 1 2 3 4 5],
  [0 1 2 3 4 5], [0 1 2 3 2 5], [0 1 2 2 2 5], [0 1 2 2 2 5],
  [0 1 2 2 2 5], [0 1 2 2 2 5], [0 1 2 2 2 5], [0 1 2 2 2 5],
  [0 1 2 2 2 5], [0 1 2 2 2 5] | [0 1 2 2 2 5]"
```

---

## C.2 RESULTS - LLAMA-3.2-1B

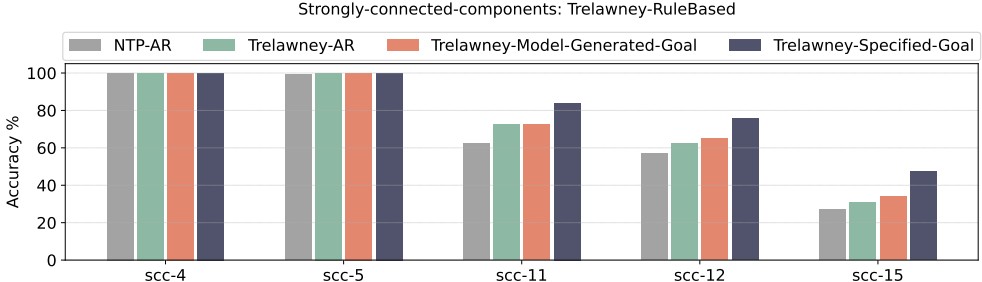

Figure 7: Accuracies - Strongly connected components TRELAWNEY-Rule-Based

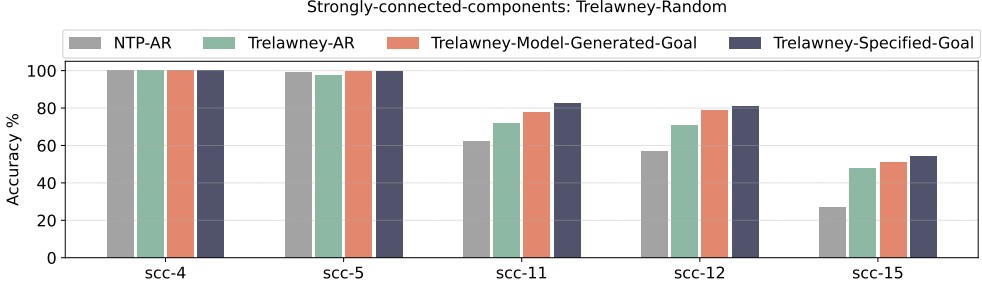

Figure 8: Accuracies - Strongly connected components TRELAWNEY-Random

## D    ZEBRA PUZZLES

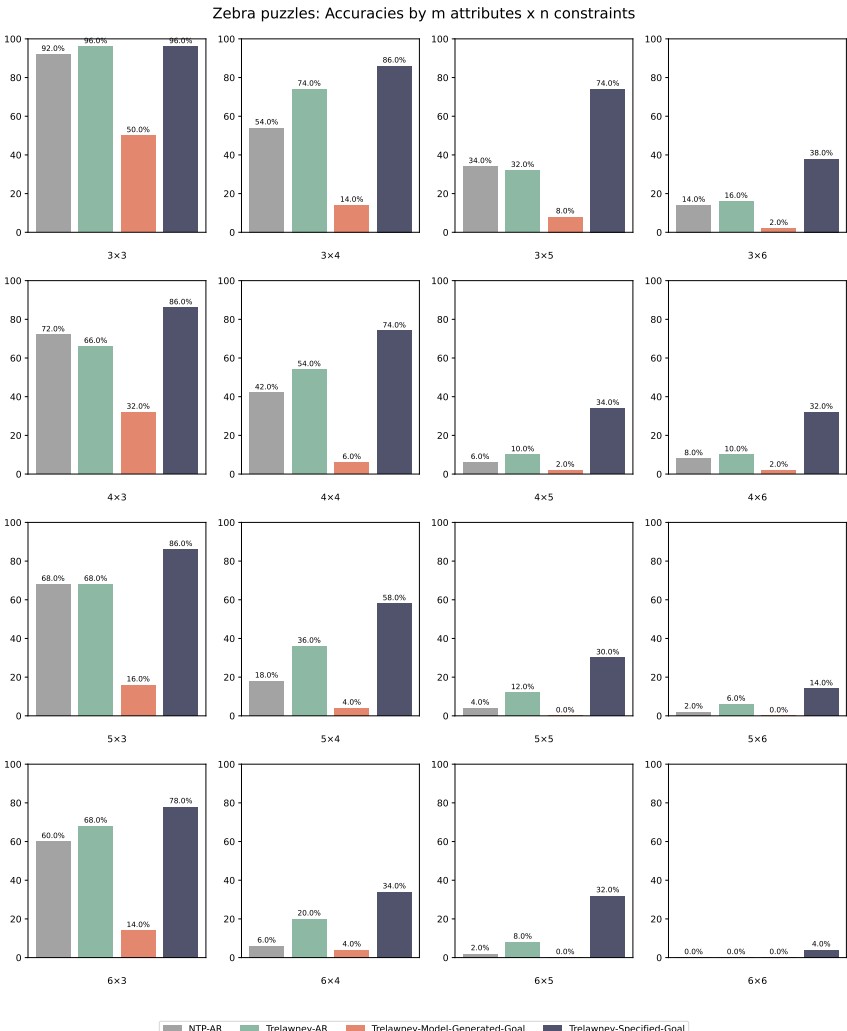

Figure 9: Accuracies by zebra puzzle complexity

| | (3,3) | (3,4) | (3,5) | (3,6) | (4,3) | (4,4) | (4,5) | (4,6) | (5,3) | (5,4) | (5,5) | (5,6) | (6,3) | (6,4) | (6,5) | (6,6) | Total |
|---|---|---|---|---|---|---|---|---|---|---|---|---|---|---|---|---|---|
| NTP | 46/50 | 27/50 | 17/50 | 7/50 | 36/50 | 21/50 | 3/50 | 4/50 | 34/50 | 9/50 | 2/50 | 1/50 | 30/50 | 3/50 | 1/50 | 0/50 | 241/800 |
| TRELAWNEY-autoreg. | 48/50 | 37/50 | 16/50 | 8/50 | 33/50 | 27/50 | 5/50 | 3/50 | 34/50 | 18/50 | 6/50 | 3/50 | 34/50 | 10/50 | 4/50 | 0/50 | 286/800 |
| TRELAWNEY-spec. | 48/50 | 43/50 | 37/50 | 19/50 | 43/50 | 37/50 | 17/50 | 16/50 | 43/50 | 29/50 | 15/50 | 7/50 | 39/50 | 17/50 | 16/50 | 2/50 | 428/800 |
| TRELAWNEY-gen. | 25/50 | 7/50 | 4/50 | 1/50 | 16/50 | 3/50 | 1/50 | 1/50 | 8/50 | 2/50 | 0/50 | 0/50 | 7/50 | 2/50 | 0/50 | 0/50 | 77/800 |

Table 6: Zebra puzzles accuracy by problem complexity. Number of m x n puzzles solved, where m is number of attributes and n is the number of constraints. Trelawney autoregressive outperforms next token prediction on more complex tasks. More constraints often lead to longer traces and chains, which alludes to better long horizon planning. On model generated future goals, our method does not show significant performance gains - by nature of the constraint satisfaction problem, it is inherently hard to find good future branching points.

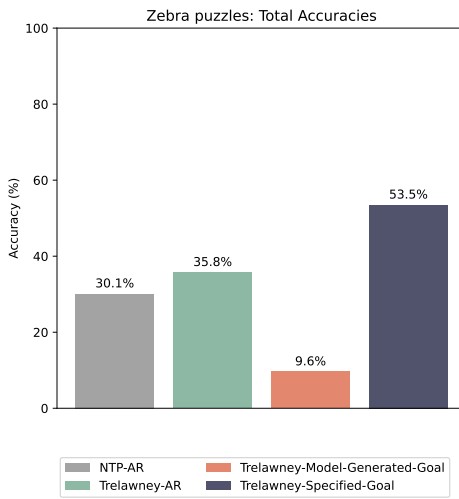

Figure 10: Our zebra puzzle evaluation demonstrates improved performance on autoregressive generation. On model generated future goals, our method does not show significant performance gains - by nature of the constraint satisfaction problem, it is inherently hard to find good future branching points.

# E  STORY GENERATION

## E.1  EXAMPLES

Once upon a time, there was a little girl named Lily. Lily loved to play in the rain. She would jump in all the puddles and get very wet. One day, Lily saw a cute little frog hopping around in the rain. "Hello little frog! What are you doing in the rain?" asked Lily. "I like the rain because it helps create more puddles for me to swim in," replied the frog. Lily smiled and said, "That's so cool! I love playing in the rain too." …

Once upon a time, there was a little girl named Lily. Lily loved to play in the rain. **<T>** I think **[3]** sentences away should be:**"Hello little frog! </T>** **[1]** She would jump in all the puddles and get very wet. **[2]** One day, Lily saw a cute little frog hopping around in the rain. **[3]** **"Hello little frog! [4]** What are you doing in the rain?" asked Lily. **[5]** "I like the rain because it helps create more puddles for me to swim in," **[6]** replied the frog. **[7]** Lily smiled and said, **[8]** "That's so cool! **[9]** I love playing in the rain too." …

Once upon a time, there was a little girl named Lily. Lily loved to play in the rain. **<T>** I think **[3]** sentences away should be:**"Hello little frog! </T>** She would jump in all the puddles and get very wet. One day, Lily saw a cute little frog hopping around in the rain. **"Hello little frog!** What are you doing in the rain?" asked Lily. "I like the rain because it helps create more puddles for me to swim in," replied the frog. Lily smiled and said, "That's so cool! I love playing in the rain too." …

Baseline                Trelawney - Explicit Position          Trelawney - Implicit Position

Figure 11: Construction of $\mathcal{D}'$

## E.2  GPT4-AS-JUDGE

**Unconditional generation.**  System prompt used and an example

### Unconditional generation

**System prompt:** You are evaluating two story generation models. First, analyze the outputs and note their properties. Then, make an argument for why one model is better than the other, or state that both models perform similarly. Finally, Respond with WINNER: Story 1/2 (depending on which is better) or respond with DRAW if both are equal.

**User**

```
Story 1:
"Once upon a time, there was a naughty bee named Buzzy.Buzzy
↪   loved to fly around the big tree and play with the other
↪   bees.One day, Buzzy saw a big, juicy flower and wanted to
↪   eat it.But the flower was too high up for Buzzy to
↪   reach.\n\nBuzzy had an idea!He flew up to the top of the
↪   tree and found a long stick.He used the stick to reach
↪   the flower and eat it.Buzzy was very happy and flew back
↪   down to tell the other bees about his adventure.\n\nThe
↪   other bees were impressed with Buzzy's cleverness.They
↪   all decided to try and reach the flower too.They worked
↪   together and used their wings to fly up to the top of the
↪   tree.Buzzy was proud of them and they all shared the
↪   delicious flower together.From that day on, Buzzy and the
↪   other bees were known as the best bee team in the
↪   forest.",

Story 2:
"Once upon a time, there was a naughty bee named Buzzy.Buzzy
↪   loved to fly around the big tree and play with the other
↪   bees.One day, Buzzy saw a big, juicy flower and wanted to
↪   eat it.But the flower was too high up for Buzzy to
↪   reach.\n\nBuzzy had an idea!He flew up to the top of the
↪   tree and found a long stick.He used the stick to reach
↪   the flower and eat it.Buzzy was very happy and flew back
↪   down to tell the other bees about his adventure.\n\nThe
↪   other bees were impressed with Buzzy's cleverness.They
↪   all decided to try reaching the flower too.They worked
↪   together and soon they were all able to reach the flower
↪   and eat it.Buzzy was proud of his friends and they all
↪   had a fun day playing together."
```

**Judge**

```
"Both stories have similar structures, characters, and
↪   plotlines. They both follow the adventure of a bee named
↪   Buzzy who uses a stick to reach a flower that was too
↪   high up. The other bees in both stories are impressed by
↪   Buzzy's cleverness and decide to try reaching the flower
↪   too. \n\nHowever, there are subtle differences in the way
↪   the stories are told. Story 1 ends with a stronger
↪   conclusion, stating that Buzzy and the other bees were
↪   known as the best bee team in the forest. This gives a
↪   sense of accomplishment and closure to the story. On the
↪   other hand, Story 2 ends on a more casual note, stating
↪   that they all had a fun day playing together. \n\nIn
↪   terms of language and style, Story 1 uses more
↪   descriptive language and provides more details, such as
↪   the bees using their wings to fly up to the top of the
↪   tree. Story 2 is more straightforward and uses simpler
↪   language.\n\nIn conclusion, while both stories are
↪   similar, Story 1 is slightly better due to its stronger
↪   conclusion and more descriptive language.\n\nWINNER:
↪   Story 1"
```

**Conditional generation** : We assess if the model was able to reach the specified goal and if the generated stories are coherent. We run two ablations of our method on the baseline trained model 1) Few shot prompts with instructions in natural language - the baseline is unable to reach any of the goals effectively, 2) Few shot prompts (8 examples) with goals specified as tags. Note: we do not use special tokens, only patterns resembling tags. For fairness we report the version with tags in the main paper.

**Conditional generation**

**System prompt:** You are evaluating two story generation models for goal reaching ability. The goal is specified within `<LOOKAHEAD>` and `</LOOKAHEAD>`, analyze if the stories reach the goal and Then, make an argument for why one model is better than the other, or state that both models perform similarly. Finally, Respond with WINNER: Story 1/2 (depending on which is better) or respond with DRAW if both are equal

**User**

```
Story 1:

"Once upon a time, there was a goat.The goat was outside and
↪  it was very hot.<LOOKAHEAD> I think [4] sentences away
↪  should be: The goat looked for another way inside but
↪  there was none. </LOOKAHEAD>The goat was very thirsty and
↪  it wanted to drink some water.\n\nThe goat saw a big tree
↪  and it thought it would be a good place to drink
↪  water.But when the goat got to the tree, it was too high
↪  up and the goat couldn't reach the water.\n\nThe goat
↪  looked for another way inside but there was none.So the
↪  goat had to stay outside and wait for the water to come
↪  down from the sky.\n\nFinally, the water came down and
↪  the goat was very happy.It drank the water and felt much
↪  better.The goat was very grateful for the water and it
↪  thanked the sky for providing it.",

Story 2:

"Once upon a time, there was a goat.The goat was outside and
↪  it was very hot.<LOOKAHEAD> I think [4] sentences away
↪  should be: The goat looked for another way inside but
↪  there was none. </LOOKAHEAD>Then, the goat saw a big tree
↪  and thought it would be a good place to hide.\n\nThe goat
↪  went to the tree and climbed up.It was very hot inside
↪  the tree, but the goat was happy to be safe.The goat
↪  stayed in the tree for a long time, but it was not happy
↪  because it was not able to get out.\n\nFinally, the sun
↪  went down and the goat was able to get out of the
↪  tree.The goat was very tired and hot, but it was happy to
↪  be outside again.The goat learned that sometimes it is
↪  not good to hide in a tree when it is too hot.",
```

**Judge**

```
"Both stories have the same goal: \"The goat looked for
↪  another way inside but there was none.\" Story 1 reaches
↪  this goal perfectly, as the goat tries to find another
↪  way to get water but fails, and the sentence is included
↪  verbatim. Story 2, however, does not reach the goal. The
↪  goat does look for a place to hide, but it successfully
↪  finds one in the tree, and the sentence is not included
↪  in the story.\n\nTherefore, Story 1 is better at reaching
↪  the specified goal than Story 2. Story 1 not only
↪  includes the goal sentence but also builds a coherent
↪  narrative around it, while Story 2 deviates from the goal
↪  and does not include the sentence.\n\nWINNER: Story 1"
```

### E.3 LLM-JUDGE EVALUATIONS

We use GPT-4 as the judge to evaluate stories generated by both models:

| Story Generation | Baseline | Ours | Draw |
|---|---|---|---|
| Autoregressive | $0.4526^{+0.494}_{-0.411}$ | $0.4035^{+0.445}_{-0.362}$ | $0.1439^{+0.175}_{-0.116}$ |
| Few shot natural language v. TRELAWNEY | $0.0000^{+0.006}_{-0.000}$ | $0.8860^{+0.911}_{-0.858}$ | $0.1139^{+0.142}_{-0.089}$ |
| Few shot tags v. TRELAWNEY | $0.1734^{+0.207}_{-0.144}$ | $0.7653^{+0.799}_{-0.729}$ | $0.0612^{+0.084}_{-0.043}$ |

Table 7: Tiny stories win rate with confidence intervals at 95th percentile

**Failure modes**: Often, both models are unable to reach the goal, then the judge outputs DRAW. In some generations, we note that while the full sentence may not be copied verbatim, we still have coherent generations. In implicit generations, the number of sentences away is less accurate than explicitly specifying them.

### E.4 PERPLEXITY

WikiText Perplexity on models trained with TRELAWNEY are comparable to models trained with standard next token prediction, indicating no noticeable loss in text generation abilities.

| | Bits-per-byte ($\downarrow$) | Byte-Perplexity ($\downarrow$) | Word-Perplexity ($\downarrow$) |
|---|---|---|---|
| Next-Token-Prediction | 0.6958 | 1.6198 | 13.1865 |
| TRELAWNEY | 0.6975 | 1.6217 | 13.2669 |

Table 8: Perplexity metrics on wikitext

# F PRETRAINING

## F.1 PERPLEXITY EVALUATIONS

| Model | C4 | Dolma | FalconRW | GAB | M2D2-S | M2D2-W | Mano | PTB | R.Pajama | Twitter | Wikitext | Avg |
|---|---|---|---|---|---|---|---|---|---|---|---|---|
| **Word Perplexity** | | | | | | | | | | | | |
| NTP-124M | 67.487 | 230.671 | 83.168 | 14851.327 | 108.632 | 99.483 | 192.435 | 118.147 | 1699.826 | 10240.487 | 55.138 | 2522.436 |
| TRELAWNEY-124M | 67.822 | 207.702 | 81.025 | 13819.647 | 107.473 | 95.961 | 171.096 | 111.094 | 1449.914 | 10876.189 | 54.425 | 2458.332 |
| NTP-1B | 39.338 | 105.296 | 46.063 | 5300.855 | 59.847 | 51.554 | 103.952 | 56.299 | 585.386 | 5122.568 | 28.539 | 1045.429 |
| TRELAWNEY-1B | 39.507 | 108.858 | 45.852 | 5405.941 | 60.093 | 51.409 | 106.616 | 53.839 | 735.245 | 5321.214 | 28.541 | 1087.011 |
| **Byte Perplexity** | | | | | | | | | | | | |
| NTP-124M | 2.018 | 2.309 | 2.103 | 3.644 | 2.272 | 2.097 | 2.599 | 2.303 | 2.808 | 5.300 | 2.116 | 2.688 |
| TRELAWNEY-124M | 2.019 | 2.271 | 2.094 | 3.609 | 2.267 | 2.085 | 2.544 | 2.279 | 2.747 | 5.358 | 2.111 | 2.671 |
| NTP-1B | 1.844 | 2.047 | 2.047 | 3.172 | 2.047 | 1.886 | 2.324 | 2.023 | 2.422 | 4.677 | 1.871 | 2.383 |
| TRELAWNEY-1B | 1.845 | 2.057 | 1.903 | 3.181 | 2.048 | 1.886 | 2.335 | 2.008 | 2.500 | 4.709 | 1.871 | 2.395 |
| **Bits per Byte** | | | | | | | | | | | | |
| NTP-124M | 1.013 | 1.207 | 1.073 | 1.866 | 1.184 | 1.068 | 1.378 | 1.204 | 1.490 | 2.406 | 1.081 | 1.361 |
| TRELAWNEY-124M | 1.041 | 1.183 | 1.066 | 1.852 | 1.181 | 1.060 | 1.347 | 1.188 | 1.458 | 2.422 | 1.078 | 1.350 |
| NTP-1B | 0.883 | 1.033 | 0.929 | 1.666 | 1.033 | 0.916 | 1.217 | 1.017 | 1.276 | 2.226 | 0.904 | 1.191 |
| TRELAWNEY-1B | 0.884 | 1.041 | 0.928 | 1.669 | 1.034 | 0.915 | 1.223 | 1.005 | 1.322 | 2.235 | 0.904 | 1.196 |

Table 9: Perplexity and compression metrics across multiple datasets from the Paloma suite.

## F.2 STORY GENERATION

| | | NTP | TRELAWNEY | Draw |
|---|---|---|---|---|
| AutoReg. | 124M | $0.461^{+0.50}_{-0.42}$ | $0.45^{+0.49}_{-0.40}$ | $0.088^{+0.11}_{-0.06}$ |
| | 1B | $0.436^{+0.47}_{-0.39}$ | $0.44^{+0.47}_{-0.39}$ | $0.123^{+0.15}_{-0.09}$ |
| Few-shot | 124M | $0.085^{+0.11}_{-0.06}$ | $0.761^{+.79}_{-.72}$ | $0.153^{+0.18}_{-0.12}$ |
| | 1B | $0.036^{+0.05}_{-0.02}$ | $0.866^{+0.89}_{-0.83}$ | $0.096^{+0.12}_{-0.07}$ |

Table 10: Win rate with confidence intervals at 95th percentile for conditional generation without task specific finetuning on models pretrained with Trelawney and NTP

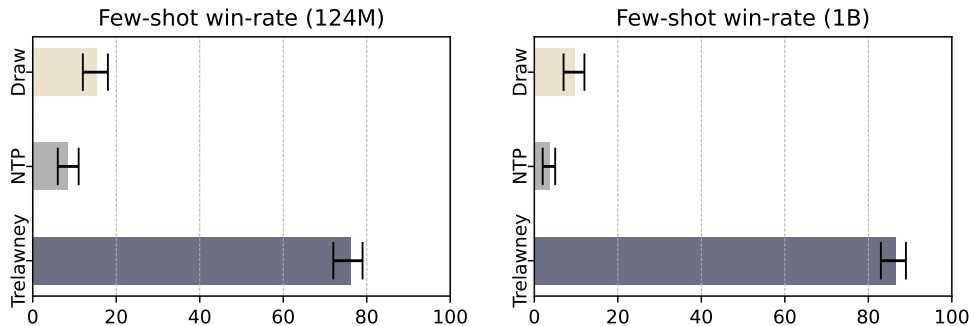

(a) In few shot evaluations for goal conditioned generation, GPT-4 prefers Trelawney trained models for goal conditioned behavior.

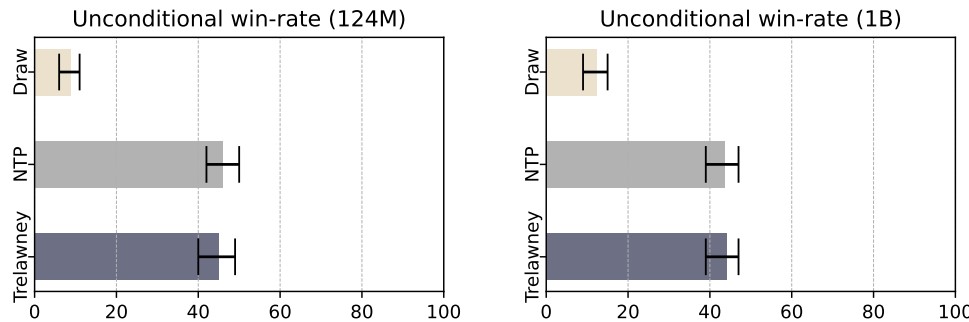

(b) Across model sizes of 124M (left) and 1B (right), there is little to no degradation in coherence and clarity in unconditional story generation for models trained with Trelawney.

