# OpenReview forum: "Looking beyond the next token"
_ICLR.cc/2026/Conference — Submitted to ICLR 2026_

### Official Review · Reviewer_t56p · 2025-10-26

**Soundness:** 3
**Presentation:** 3
**Contribution:** 3
**Rating:** 6
**Confidence:** 4

**Summary:**

This paper introduces TRELAWNEY, a data-centric method designed to enhance the planning capabilities of autoregressive language models without altering the model architecture or training infrastructure. The core idea is to augment the training data by interleaving original sequences with special lookahead tokens that contain information about future parts of the sequence. The motivation stems from the limitations of the standard next-token prediction (NTP) objective, which struggles with tasks requiring long-range dependencies or non-linear reasoning, often failing to learn crucial early decision points due to teacher forcing. By exposing the model to future sub-goals or segments during training, TRELAWNEY aims to improve performance on tasks like path planning, algorithmic reasoning, and controllable generation, and potentially enable models to generate their own long-term goals.

**Strengths:**

- The paper proposes an interesting and notably simple approach to potentially improve LLM planning abilities solely through data augmentation. Modifying the data factorization to include future information is an intuitive way to address known limitations of autoregressive NTP.

- The motivation is well-presented and easy to understand. The discussion of NTP's drawbacks (like the "Clever Hans Cheat" and "Indecipherable Token Problem") effectively highlights why alternative training signals might be beneficial.

- The method requires no changes to the underlying model architecture or complex training objectives, making it potentially easy to integrate into existing pre-training or fine-tuning pipelines.

**Weaknesses:**

- The primary concern revolves around the method's scalability and applicability to more general and complex tasks beyond the specific benchmarks tested. While the results on path planning, algorithmic reasoning (SCC), Zebra puzzles, and the relatively simple TinyStories generation task are encouraging, these are still quite structured or constrained domains.

- It remains unclear whether this data augmentation strategy would be effective for tasks requiring more sophisticated reasoning, such as complex mathematical problem-solving or nuanced long-form text generation, where identifying meaningful "future" segments to insert might be non-trivial or less impactful.

- Since the paper's contribution is almost entirely empirical, resting on the performance gains shown in these specific tasks, the lack of strong evidence for broader applicability makes the overall impact uncertain.

**Questions:**

- Could the authors elaborate on how the TRELAWNEY augmentation strategy might be applied to more general reasoning tasks, such as solving mathematical problems? What kind of "future" information (e.g., intermediate steps, final answer structure) could be inserted, and would it provide a meaningful signal?

- In the story generation task (Section 4.4), the specific format used for the lookahead token is "I want the [k]-th sentence from here to be [s]". This phrasing seems quite specific to writing tasks. Is there a particular reason for this choice? Does this specific, natural language phrasing risk overfitting to the TinyStories dataset structure or this particular type of instruction, potentially limiting generalization?

Regarding the experimental details:

- In Figure 5 and Table 2 (Appendix), what does the "Draw" category represent in the GPT-4 evaluation? (Presumably, cases where GPT-4 rated the TRELAWNEY and baseline outputs as equal)
- In Table 2, what exactly does the "Few-shot" condition entail?

---

> ### Author Response · Authors · 2025-11-21
>
> We thank the reviewer for their thoughtful comments!
>
> > The primary concern revolves around the method's scalability and applicability to more general and complex tasks beyond the specific benchmarks tested
>
> Your concern is well taken, and we will make it clearer in the updated manuscript that the key results are on pretraining of 124M and 1B scale models, and we will also expand on the set of presented results. Our  simple tasks are designed as diagnostic to demonstrate the specific implications of our method in controllable settings. Identifying meaningful future segments in challenging domains is non-trivial, we will provide some directions for mathematical reasoning and code. On scalability and impact - recent work from an industry lab (to be cited in the camera-ready) uses a related future-aware objective at 3B/8B and reports gains on math and reasoning benchmarks, providing independent evidence that predicting informative futures is a useful inductive bias. We omit the full reference here to preserve anonymity and will include it in the camera-ready; we apologize for the inconvenience and appreciate your understanding.
>
>
> > Could the authors elaborate on how the TRELAWNEY augmentation strategy might be applied to more general reasoning tasks, such as solving mathematical problems? What kind of "future" information (e.g., intermediate steps, final answer structure) could be inserted, and would it provide a meaningful signal?
>
> For more structured domains such as formal theorem proving (e.g., LEAN), futures could consist of lemmas, intermediate goals, or proof outlines interleaved with the raw proof steps. In code-like environments, futures could be function signatures, docstrings, or high-level comments that sketch the intended behavior.
> More generally, when humans write code or proofs they plan hierarchically; TRELAWNEY can use these intermediate plans as future segments, providing meaningful signals for long-horizon reasoning. In essence, your question captures the importance of this shift in perspective, recognizing how general the notion of future goal really is despite largely being elided by the literature.
>
> > In the story generation task (Section 4.4), the specific format used for the lookahead token is "I want the [k]-th sentence from here to be [s]". This phrasing seems quite specific to writing tasks. Is there a particular reason for this choice? Does this specific, natural language phrasing risk overfitting to the TinyStories dataset structure or this particular type of instruction, potentially limiting generalization?
>
> The guiding principle in this phrasing is to provide 1) a notion of relative position to the model – ie how far ahead 2) the goal – ie what should you use/ expect to see. This can easily be modified to a more general phrasing or adapted with more concise/general statements - since we are in the text domain for pretraining, we chose to retain this phrasing.
>
>
> > Regarding the experimental details: In Figure 5 and Table 2 (Appendix), what does the "Draw" category represent in the GPT-4 evaluation? (Presumably, cases where GPT-4 rated the TRELAWNEY and baseline outputs as equal) In Table 2, what exactly does the "Few-shot" condition entail?
>
> * DRAW has two concrete meanings in our GPT-4 evaluations.
>     * In the goal-conditioned setting, a draw is recorded when GPT-4 judges that both models fail to satisfy the stated goal (e.g., neither story respects the constraint).
>     * In the unconditional setting, a draw occurs when, over the 6 pairwise comparisons for a prompt, GPT-4 prefers each model the same number of times, so there is no clear winner.
> * Few shot: We provide the model k examples and prompt it to complete the next one.

---

### Official Review · Reviewer_vA7g · 2025-10-29

**Soundness:** 2
**Presentation:** 3
**Contribution:** 2
**Rating:** 2
**Confidence:** 5

**Summary:**

This paper introduces a data augmentation procedure called Trelawney which aims to make models better at learning long-range dependency information. This is an effort to address previously identified issues with in models trained auto-regressively with teacher forcing to solve certain kinds of planning tasks. During training Trelawney injects information from later in the sequence into the context, between two special tokens. This aims to have the model learn to represent information about the future earlier during generation - the authors make an analogy between this and planning. Experiments show this approach can improve performance on 3 different toy tasks aimed at assessing this kind of goal-directed planning. Additionally authors show when leveraged during pre-training this augmentation doesn't impair language modelling performance and can improve a particular kind of controllability.

**Strengths:**

Overall the Paper is relatively well written, and the results are clearly presented. Additionally the approach introduced is general and can be applied off the shelf to existing decoder-only models with only a modification to include the special tokens indicating when future information is presented. The authors make an effort to show results on small tasks as well as pre-training. While the models are relatively small a 1B model seems sufficient to show the approach can scale.

**Weaknesses:**

Overall, the main weaknesses appear to be in terms of the paper's novelty and evaluation. To highlight two parts of the evaluation that are concerning:

First, the results in table 1 show that inserting random future sequences (z) is more effective than inserting sequences that are deliberately selected to be relevant to the goal. The authors don't offer a compelling explanation for this, despite the fact that it would appear to undermine their argument that what makes their approach work is the inclusion of goal-relevant information in z. Additionally the condition that comprises the last two rows of table 1 is one where during inference the model is given part of the solution as a "user specified" z. It should not be surprising that this improves performance, and the authors should make that clear.

Second. The evaluation in section 4.4 relies on using GPT4 to evaluate how well a model generates a story according to a goal like "I want the 4th sentence from here to be "Hello little frog."" They show that models trained with Trelawney are better at complying with these explicitly stated goals than models trained without explicit goal supervision. This appears to be done by injecting a "goal" subsequence z during generation for both the Trelawney model and the baseline (according to the example shown in the appendix). This goal injection is exactly the Trelawney training objective used to train the author's model and not the baseline. Therefore these results show if you train on an objective, you will do better on it during evaluation --- a finding that lacks novelty. By contrast the results in table 1 do not suffer from this - they show training on the Trelawney objective improves performance on a task objective. In section 4.4 the task objective appears to be almost indistinguishable from the Trelawney objective. I have similar concerns for the "conditional generation" results in section 4.5 which the authors say follows the same protocol used in 4.4.


As some minor points: the authors appear to conflate next token prediction and teacher forcing. They refer to baselines as "NTP" despite the fact that the objective with Trelawney is still next token prediction just with augmented data. I may have missed it but I don't think the authors introduce the different between the implicit and explicit condition before referring to it.

**Questions:**

Why does the random condition out perform fixed in table 1?

In the Path Planning results why does Trelawney perform better on higher degrees than it does on longer paths?

Is the "spec." condition in table 1 just giving part of the solution to the model - if so what should we conclude from an increase in performance?

Does Trelawney pre-training improve instruction following on any other tasks that do not explicitly resemble the Trelawney objective?

At line 048 in the introduction you state your approach "embed[s] inductive biases directly." What inductive biases does your approach embed?

---

> ### Author Response · Authors · 2025-11-21
>
> We thank the reviewer for their comments and provide clarifications below.
>
> > First, the results in table 1 show that inserting random future sequences (z) is more effective than inserting sequences that are deliberately selected to be relevant to the goal.
>
>
> Thank you for pointing out this potential confusion! For the evaluation, our hypothesis is that random augmentation acts as a regularizer relative to user-specified goals because it effectively increases data diversity. We will better highlight this in the paper to make this point clearer and more prominent. Second, even a randomly chosen future state is still a future state on the trajectory that the model must generate, so it is fully consistent with our motivation that it is beneficial to be explicitly aware of possible future states rather than only the immediate next token. We elaborate further on this in our responses to your later questions.
>
>
> > Therefore these results show if you train on an objective, you will do better on it during evaluation --- a finding that lacks novelty.
>
>
> Regarding Section 4.4, we respectfully disagree that evaluating on a metric aligned with the training objective diminishes novelty. It is standard practice, for example, to evaluate next-token pretraining with perplexity, which exactly matches the training objective, especially when the paper also demonstrates additional, nontrivial properties of that objective. In our case, it would be more concerning if the model failed to improve on this evaluation. Moreover, for the experiments with our pretrained model, we did not train on the tiny story corpus, so this evaluation is out-of-distribution and indicates that the learned behavior generalizes beyond the training data. Our longer-term vision is that this behavior can ultimately support new capabilities in the post-training stage, although we do not explore this direction in the current work.
> To summarize, we show that our objective (1) has interesting regularizing effects that are not obviously related to the objective itself, (2) does not interfere with standard language modeling, and (3) successfully induces behaviors that are closely aligned with the objective  (i.e., indicating that  the training is effective).
>
>
> > Why does the random condition out perform fixed in table 1?
>
> Thank you for the question. In most settings, it is difficult to come up with domain specific augmentation, so the fact it works better is a great benefit for the generality of the method. Note that random conditioning is still conditioning on a plausible future position (just not necessarily the one we are evaluating on) so there is no contradiction between our argument. It is natural to expect that user specific conditioning would work better than random. This was our original hypothesis too so it was surprising that random augmentation was better on a task that it was not specifically trained for. Our working hypothesis is that random samples have different future Zs after the decision node on each example, so the model sees a wide variety of subgoals and path lengths. This induces a regularizing effect that prevents overfitting to a single “canonical” subpath and we hypothesize that it yields a representation at the decision point that is more robust to where along the path the future is taken.
>
> > In the Path Planning results why does Trelawney perform better on higher degrees than it does on longer paths?
>
> The path planning example - higher degree denotes higher branching factor at a specific point, longer length denotes longer horizon planning. Since we have a single point of failure, if the goal is close enough - the model is able to learn. Further -  larger models learn longer paths - 3B model solves the longest graph effectively whereas the 1B model does not (see appendix), indicating that  it is a model capacity issue. This observation is not unique to this work. Many other works that use this task observe similar issues.

---

> > ### Author Response · Authors · 2025-11-21
> >
> > > Is the "spec." condition in table 1 just giving part of the solution to the model - if so what should we conclude from an increase in performance?
> >
> > Yes, the “Spec.” condition provides part of the trajectory to the solution in the form of a user-specified future z. We never give the exact solution in z.
> > * Gen: can the model generate a useful goal and then follow it?
> > * Spec: given an externally provided goal, can the model use it effectively?
> >
> > We expect Spec $\geq$ Gen in performance, since failures in Gen can come from either poor goal generation or poor goal use, whereas Spec isolates the latter. The gain in the Spec condition shows that TRELAWNEY can reliably execute provided goals (controllability), and is  not “giving away” the answer. Additionally, it lets us specify concrete goals. (e.g. in creative story generation we can fix particular plot points in advance.)
> >
> >
> > > Does Trelawney pre-training improve instruction following on any other tasks that do not explicitly resemble the Trelawney objective?
> >
> > Our experiments are at 124M and 1B scales, due to limited compute. At this scale, models cannot do meaningful instruction following. Trelawney matches or slightly improves over NTP on these tasks, indicating that the auxiliary future supervision does not harm and can modestly help generic pretraining evaluations, even when no futures are used at test time.
> >
> > > At line 048 in the introduction you state your approach "embed[s] inductive biases directly." What inductive biases does your approach embed?
> >
> > Our approach embeds an inductive bias toward future-aware representations: at decision states, the model is trained to predict a block of future tokens Zs, so these states encode information about long-horizon outcomes rather than only the immediate next token.

---

### Official Review · Reviewer_cJtC · 2025-10-30

**Soundness:** 3
**Presentation:** 3
**Contribution:** 2
**Rating:** 4
**Confidence:** 3

**Summary:**

The paper aims to improve the conventional next-token prediction paradigm in language model training, which restricts models to predict each token solely based on past context. It argues that humans generate language with awareness of future goals rather than relying only on previous information. To address this limitation, the authors propose TRELAWNEY, a simple data augmentation approach that inserts special tokens $\langle T\rangle \ldots \langle \/T\rangle$ containing future text segments representing goals or upcoming content into training sequences. Using this method, models are trained on a mixture of original and augmented data. Experiments on synthetic planning tasks, algorithmic reasoning, and story generation demonstrate that TRELAWNEY enhances performance on tasks requiring foresight without degrading standard language modeling ability.

**Strengths:**

Please find the strengths below:
1. The paper enhances the foresight capability of language models by introducing explicit future markers $\langle T\rangle \ldots \langle \/T\rangle$ instead of modifying the model architecture as in prior approaches.
2. The experiments cover multiple dimensions, including synthetic planning tasks (star-graph), algorithmic reasoning tasks (strongly connected components), and natural language story generation, providing thorough and diverse validation as well as comparisons across different augmentation strategies.
3. The paper is clearly written, with well-motivated problem statements and detailed, reproducible descriptions of the proposed method.

**Weaknesses:**

Please find the weaknesses below:
1. The experiments are diverse across different types of datasets but are conducted on relatively small models and mostly synthetic or simplified data.
2. The insertion of future tokens $\langle T\rangle \ldots \langle /T\rangle$ and the choice of where to place them rely on manually defined or heuristic rules. There is no theoretical guidance on how to modify the training data effectively.
3. The connection between the proposed augmentation and established notions such as planning, goal inference, or information flow is discussed only at an intuitive level.
4. For open-ended or creative text generation, it remains unclear how the insertion of future tokens can be meaningfully applied.

**Questions:**

The questions are related to the weaknesses:
1. Is it possible to provide any theoretical guidance or justification for the insertion of future tokens and the choice of where to place them?
2. Is it possible to conduct a more rigorous analysis of the connection between the proposed augmentation and established notions such as planning, goal inference, or information flow?
3. Since the experiments only involve small-scale fine-tuning on a single model, is it possible to extend the evaluation to more advanced and larger language models?

---

> ### Author Response · Authors · 2025-11-21
>
> > The experiments are diverse across different types of datasets but are conducted on relatively small models and mostly synthetic or simplified data.
>
> Thank you for sharing this concern, we will be sure to clarify why each experimental domain was chosen in the updated manuscript. In short, we chose the domains explicitly to ensure we could clearly isolate different factors in each experiment (e.g. branching, entropy, vocabulary, etc).  The synthetic domains are used for fine-tuning so we can characterize behavior, while all pretraining experiments are obviously conducted on general text data.  However, without access to commercial compute, we are restricted to pretraining at the 1B scale.
> This choice of experiments allows our manuscript to both substantiate our claims and suggest potential additional benefits at larger scales. In addition, recent work from an industry lab (to be cited in the camera-ready) uses a related future-aware objective at 3B/8B and reports gains on math and reasoning benchmarks, providing independent evidence that predicting informative futures is a useful inductive bias. We omit the full reference here to preserve anonymity and will include it in the camera-ready; we apologize for the inconvenience and appreciate your understanding.
>
> > The insertion of future tokens  and the choice of where to place them rely on manually defined or heuristic rules. There is no theoretical guidance on how to modify the training data effectively.
>
>
> Thank you for raising this potential confusion, the core results are all with  heuristic rule, that follow a clear principle: insert futures at *high-uncertainty / high-entropy* decision points where multiple long-horizon branches are still viable. This is the intent of introducing the didactic path planning task.
> Manual rules are only used in synthetic settings, to demonstrate the importance of bottlenecks.  Clearly, only in controlled experiments are these easily identifiable from the known structure, so entropy-based placement is straightforward. In general language modelling, tokenization and context window make direct entropy estimation much noisier and less reliable, which is why we rely on simpler, data-driven heuristics there.
>
> > For open-ended or creative text generation, it remains unclear how the insertion of future tokens can be meaningfully applied.
>
> Our story experiments are intended as an initial demonstration in an open-ended setting: futures specify soft narrative goals (e.g., “the 8th sentence resolves X”), while the rest of the story is unconstrained. In practice, we anticipate that futures can be authored by a user or another model, and inserted at any point to steer global structure (plot turns, style shifts, safety constraints) without changing the base architecture or decoding algorithm. We will extend the discussion and examples in §4 to make these use cases clearer and to better articulate how future tokens serve as a general interface for high-level control in creative text generation.

---

> > ### Author Response · Authors · 2025-11-21
> >
> > > Is it possible to provide any theoretical guidance or justification for the insertion of future tokens and the choice of where to place them?
> >
> >
> > Absolutely! Due to space constraints, we focused on a more intuitive explanation for general audiences rather than a formal specification and justification. While a fully general theoretical treatment would be very interesting, it appears feasible only in highly simplified, toy settings that abstract away most of the phenomena we care about in practice. In this work we therefore focus on an empirical study on realistic models and tasks, and leave a more complete theoretical analysis to future work.
> >
> > We attempt to provide a proof sketch for the didactic path star task.
> >
> > Proposition : Consider the path–star environment where each sample consists of a start–goal pair $(S,G)$ and the unique shortest path \(Z\) between them, and let $H_d$ be the model state at the decision point. Let $L_{\mathrm{NTP}}$ be the standard next-token loss and define
> > $$ L_{\mathrm{aug}} = L_{\mathrm{NTP}} +
> > \lambda\,\mathbb{E}\big[-\log q_\theta(Z \mid H_d)\big], \qquad \lambda > 0.$$
> >
> > Then:
> >
> > * There exists a sequence of parameter settings $\{\theta_k\}$ with $L_{\mathrm{NTP}}(\theta_k) \to 0$ and $I_{\theta_k}(H_d; Z) = 0; \forall k$.
> > * For any global minimizer $\theta^\star$ of $L_{\mathrm{aug}}$, we have $I_{\theta^\star}(H_d; Z) = H(Z)$.
> >
> >
> > Proof Sketch:
> >
> > * Because the model class is universal, there exists a sequence ${\theta_k\}$ whose next-token conditionals converge to the data distribution $p(Z_t \mid Z_{\< t}, S, G)$, so $L_{\mathrm{NTP}}(\theta_k) \to 0$. We can choose these $\theta_k$ so that $H_d$ is a constant independent of $(S,G)$ (the predictions depend only on the visible prefix, not on $H_d$). Then, $H_d \perp Z \ \forall \ k$, hence $I_{\theta_k}(H_d; Z) = 0$.
> >
> >
> >
> > * For any $\theta$, the auxiliary term $\mathbb{E}[-\log q_\theta(Z \mid H_d)]$ is minimized iff. $q_\theta(Z \mid H_d = p(Z \mid H_d)$ almost surely. In the path–star environment, $Z$ is a deterministic function of $(S,G)$. So at any global minimizer $\theta^\star$, the conditional $p(Z \mid H_d)$ must be a point mass on the true path (otherwise the loss would be strictly larger). Thus $H(Z \mid H_d) = 0$ and therefore $I_{\theta^\star}(H_d; Z) = H(Z) - H(Z \mid H_d)= H(Z)$
> >
> >
> > > Is it possible to conduct a more rigorous analysis of the connection between the proposed augmentation and established notions such as planning, goal inference, or information flow?
> >
> >
> > Yes, in fact your first question was prescient in this regard. The same information-theoretic formalism - treating the copied future $z$ as a latent plan/goal variable and analyzing how the auxiliary loss increases $I(H_d; Z)$ - provides the more rigorous connection to planning, goal inference, and information flow.
> >
> > > Since the experiments only involve small-scale fine-tuning on a single model, is it possible to extend the evaluation to more advanced and larger language models?
> >
> > Our experiments are not limited to fine-tuning on a single model. -  Please refer to the appendix (A.5) for ablations on model sizes [0.5B, 1B, 3B scales] and autoregressive model families [state space models (see A.4) and transformers], as well as pretraining experiments at 124M and 1B scales.
> >
> > The goal in this paper is to introduce and analyze a data-centric method rather than to compete on absolute SOTA. We therefore focused on small–medium models where we could run many controlled ablations across diverse settings (graphs, algorithms, puzzles, and open-ended text). As shown in appendix A.5, our method shows improvements in long horizon planning with increase in model sizes, ie.  while 1B models are unable to implicitly solve the longest graph, 3B models are able to do so. Consistent with this observation, Wu et. al $^{[1]}$ also observe that longer horizon planning capacity increases with model size. The focus of this work is on the limitations of simple supervised learning (next token prediction) objective, not post-trained models.
> >
> > We also show that TRELAWNEY can be applied during pretraining on natural text without hurting standard downstream metrics. Scaling to 3B+ models or full-production training runs is a matter of compute, and we see this as complementary future work rather than a conceptual limitation; we will clarify this positioning in the revised version. It is encouraging to see recent work on future aware pretraining at 3B and 8B scales find similar objectives to be useful.
> >
> > [1] Wu, Wilson, John X. Morris, and Lionel Levine. "Do language models plan ahead for future tokens?." arXiv preprint arXiv:2404.00859 (2024).

---

### Official Review · Reviewer_5D2n · 2025-10-30

**Soundness:** 3
**Presentation:** 4
**Contribution:** 3
**Rating:** 6
**Confidence:** 4

**Summary:**

This paper presents a method for data manipulation in training a LLM to include specific “lookahead” fragments. The idea is that by training the model to make some longer distance predictions, it can exhibit more reasoned planning behaviour in its next-token prediction to reach these goals. This is demonstrated to be helpful in various settings, from path planning to puzzles to short story generation.

Overall the paper is interesting, the method simple but seemingly effective, and overall well evaluated (but see my many questions.) The approach may be overly simple, in comparison to other papers in this space attempting a more challenging and general problem of intra-token thinking.

**Strengths:**

The method is very simple, which adds to its appeal as it can most likely be adapted to other situations.

The paper is very well written, and the ideas clearly motivated and presented in an easily accessible manner.

The evaluation is thorough, hitting a range of very different tasks. The diversity of tasks employed helps to demonstrate the generality of the method, and the results are uniformly strong.

**Weaknesses:**

The method is very heuristic, through the manner of thinking supervision predicting the [nth] following sentence/token, rather than something more abstract or learnable from data. Models do learn to solve complex generation problems, and while some of these capabilities are explicitly trained, many emerge from next-token prediction pre-training or from reinforcement learning, e.g., of thinking models.

Related to the above, the paper neglects to cite “quiet-star” [1] which includes thinking between tokens during training, but fit using RL training. This might be learning a similar type of lookahead, but based on their paper, it looks to be much deeper reasoning and other thinking behaviours. Quiet-star is much more ambitious than this work, and I was left a bit underwhelmed by this paper as a consequence. However, I would be very interested to see whether a) empirically you can show an improvement over quiet-star on your evaluation sets, b) you can compare what’s learned in your method vs theirs, or c) you could cook something up to combine both methods, e.g., initialise quiet-star with your heuristic, and then learn more general thinking behaviour.

Some experimental aspects of the paper were unclear, see questions below. The pre-training section was particularly unclear, and I’m not sure how much this adds to the paper.

[1] Quiet-STaR: Language Models Can Teach Themselves to Think Before Speaking; Eric Zelikman et al, 2024

**Questions:**

1. See weakness above, I’d appreciate a detailed comparison to quiet-star.
1 .What are Zebra puzzles? What do the colours mean in Fig 1? Note that this figure is truncated on the right, the whole story is not visible.
1. How does supervising special “futures” based on copying future tokens resolve exposure bias / teacher forcing? This part of the narrative is not made explicit.
1. The method for adding a long sentence (line 156) between tokens looks very expensive computationally, likely requires an instruction tuned model to benefit, and it blows out the sequence length. Can you discuss these and other limitations?
1. Masking <T> generation – if this were not masked, then the model would be free to include this style of thinking during generation. Can you comment on whether this might be useful, and how this might compare to dynamic thinking (or perhaps the runtime consequences would be serious?)
1. Path Planning (Fig 3) – are the graphs provided in the dataset, and the start/goal nodes randomly selected?
1. When generating from the model, what happens if no <T> token is supplied, i.e., the generation does not include any thinking at inference time? As I understand it, all evals were run with thinking on.
1. For the story generation problem, can commercial LLMs generate stories that honor the <T> constraint, e.g., 8th sentence is X as per Figure 4? I wonder if this is within the capabilities of current models.
1. 413 “stories are shuffled” – I don’t follow why this is done, and is this at word or sentence level? Either way, it appears to corrupt the dataset to the extent that it may be useless.
1. 453 did the evaluation of pre-trained models use <T> decoding? I don’t know how to interpret the numbers in Table 2, can you clarify if they are perplexity results, and if do, does this show your method is considerably worse than Draw?

# Comments on prose:
1. 086 “exposure bias” sentence should reference on- vs off- policy learning.
1. 096 you may want to consider diffusion models

---

> ### Author Response · Authors · 2025-11-21
>
> We thank the reviewer for their comments. The goal of this paper is to introduce a simple data centric method that shows that future aware training is useful in autoregressive models. We will include the discussion of Quiet-Star in the revision among the other works that introduce architectural (MTP$^{[1]}$, Deepseek MTP$^{[2]}$, BST$^{[3]}$) or objective(QuietStar$^{[4]}$) level changes.
>
> However, we believe that the goal in this paper is complementary to Quiet-STaR. We introduce a data-centric, future-aware training augmentation that can be plugged into standard autoregressive training without any architectural changes or additional rollouts. The goal (at least not explicitly) is not to train a better reasoner but to learn a better representation. Modern model development of language models is extremely complex so we believe that the simplicity of the method is in fact ideal. Given the complexity of modern language model development, the simplicity of the method is a practical advantage.
>
> In contrast to Quiet-Star - Our approach is
> 1. FLOPs controlled and comparable to standard autoregressive training. For a sequence of length L, our objective uses exactly one forward and one backward pass of the model, identical to standard next-token prediction - because we do not require rollouts between tokens. Quiet-STaR, adds extra rollout passes for “thought” trajectories and policy-gradient updates, so its training FLOPs grow roughly in proportion to the number of thought steps per token
> 2. Can be applied in pretraining and finetuning stages.  Quiet-STaR is instantiated in a continual-training setting and relies on a modified training paradigm with policy-gradient–based updates over latent “thought” trajectories.(If introduced at pretraining stages, this may be more computationally expensive)
>
> [1] Gloeckle, Fabian, et al. "Better & faster large language models via multi-token prediction, 2024." URL https://arxiv.org/abs/2404.19737.
>
> [2] Liu, Aixin, et al. "Deepseek-v3 technical report." arXiv preprint arXiv:2412.19437 (2024).
>
> [3] Hu, Edward S., et al. "The belief state transformer." arXiv preprint arXiv:2410.23506 (2024).
>
> [4] Zelikman, Eric, et al. "Quiet-star: Language models can teach themselves to think before speaking." arXiv preprint arXiv:2403.09629 (2024).

---

> > ### Author Response · Authors · 2025-11-21
> >
> > > What are Zebra puzzles? What do the colours mean in Fig 1? Note that this figure is truncated on the right, the whole story is not visible.
> >
> >
> > We will revise Section 4.3 to more clearly introduce Zebra (Einstein) puzzles, and a small worked example in the appendix. Briefly, a Zebra puzzle is a constraint-satisfaction problem over entities (e.g., houses) and their attributes (e.g., color, nationality, pet), where the goal is to recover the unique assignment that satisfies all clues. Solving these puzzles requires reasoning over long-range dependencies and long-horizon planning in a large combinatorial search space.This makes them ideal for evaluating how well a model can work towards goals
> >
> > We will also clarify the colors in Fig. 1 (each color denotes a distinct entity/attribute value; we will add a legend) and fix the cropping so that the entire story is visible.
> >
> > > How does supervising special “futures” based on copying future tokens resolve exposure bias / teacher forcing? This part of the narrative is not made explicit.
> >
> > While attention was originally introduced as a short-cut mechanism for gradients in models, in practice current LLMs place a disproportionate amount of their representational power on local predictions, not identifying the relationships between early contexts and long-distant goals.  TRELAWNEY helps the model by modifying the training distribution so that early “decision” tokens are trained to be predictive of a future subgoal z rather than only the immediate next token. This reduces the Clever Hans effect and strengthens gradients on early high-entropy tokens, which in turn mitigates exposure bias: at inference, the model is more likely to have learned a plan that is robust to its own past predictions. We will make this causal connection explicit, and point to the gains in purely autoregressive evaluation (Table 1).
> >
> > > The method for adding a long sentence (line 156) between tokens looks very expensive computationally, likely requires an instruction tuned model to benefit, and it blows out the sequence length. Can you discuss these and other limitations?
> >
> > In principle, we agree, but in practice, our approach ends up being much shorter than CoTs and most mid-training regimes which are now standard in LLM training. However, we will update the manuscript to note this point, as we suspect there would be a cost associated with our approach when used on simpler/smaller models that do not have these enhancements.
> > To further mitigate this issue, in our experiments, we apply augmentation only to a fraction p of examples, so the effective context-length overhead is modest. Finally, we also observe benefits even without any instruction tuning, as shown by improvements in synthetic tasks and pretraining.
> >
> >
> > > Masking \<T\> generation – if this were not masked, then the model would be free to include this style of thinking during generation. Can you comment on whether this might be useful, and how this might compare to dynamic thinking (or perhaps the runtime consequences would be serious?)
> >
> > In principle, if the loss on \<T\> were not masked, the model could indeed learn to spontaneously introduce future blocks and thereby internalize this style of “thinking” during generation. We view this as especially natural in an RL or decision-making setting, where \<T\> insertion could be treated as an action and reinforced when it improves downstream reward. However, during pretraining (supervised learning), since \<T\> can occur at any position for p portion of the data. If the model were to predict \<T\>, then it would have to put non-negigible probability on generating \<T\> for many tokens, which would hurt regular text modeling and decoding. We believe a natural next step is to have a post-training phase where the model learns to leverage \<T\> spontaneously similar to dynamic thinking methods.
> >
> >
> > > Path Planning (Fig 3) – are the graphs provided in the dataset, and the start/goal nodes randomly selected?
> >
> > The graphs and the start and goal nodes are provided in the dataset.

---

> ### Author Response · Authors · 2025-11-21
>
> > When generating from the model, what happens if no \<T\> token is supplied, i.e., the generation does not include any thinking at inference time? As I understand it, all evals were run with thinking on.
>
>
> Importantly, we evaluate on standard autoregressive generation. In table 1, This is denoted as Autoreg,  standard autoregressive generation in evaluation. We share the same goal you are articulating, to demonstrate that our simple objective improves performance, even when we do not use \<T\>. We believe this supports the claim that it has learned a better representation even when the lookahead is not present.
>
>
> > For the story generation problem, can commercial LLMs generate stories that honor the [object Object] constraint, e.g., 8th sentence is X as per Figure 4? I wonder if this is within the capabilities of current models.
>
>
> For frontier models trained with RL objectives and significant posttraining,  it is possible. We hypothesize there is benefit for natively doing such training from the pretraining stage but this hypothesis cannot be verified without significant resources.
>
>
> > 413 “stories are shuffled” – I don’t follow why this is done, and is this at word or sentence level? Either way, it appears to corrupt the dataset to the extent that it may be useless.
>
> The shuffling refers only to shuffling the completed stories across trials before sending them to GPT-4 for blind evaluation, to avoid position and author bias. We never shuffle tokens or sentences within a story and do not permute the training data beyond our controlled ζ insertions, so the underlying TinyStories dataset remains intact. We will rephrase this sentence to make clear that shuffling is over evaluation order, not over the internal word/sentence order.
>
>
> > 453 did the evaluation of pre-trained models use \<T\>\</T\> decoding? I don’t know how to interpret the numbers in Table 2, can you clarify if they are perplexity results, and if do, does this show your method is considerably worse than Draw?
>
>
> We will clarify this in the updated draft -
> For the standard downstream benchmarks, the pretrained models are evaluated with plain autoregressive decoding and no \<T\> generation. The goal is to show that training with Trelawney does not interfere with normal usage of LLMs (due to not predicting \<T\>). In contrast, Table 2 reports GPT-4 pairwise preference win rates (with 95% CIs) on our conditional story-generation setup, where we do use \<T\>; “Draw” there simply means GPT-4 judged the two outputs equal in quality, not that Trelawney is worse than a separate “Draw” baseline. (See Appendix F.2 in updated draft.)
>
>
> > Comments on prose
>
>
> Thank you for pointing these out, we will be sure to update the phrasing and set of possible future directions to improve the readability and utility of paper.

---

### Meta-Review · Area_Chair_JQV3 · 2026-01-06

**Summary:**

The paper presents an interesting and simple approach, but critical weaknesses undermine its contribution. The evaluation methodology concern raised by Reviewer vA7g that models perform better on metrics aligned with their training objective represents a significant flaw that was not adequately addressed. Additionally, the counterintuitive finding that random augmentation outperforms goal-relevant augmentation lacks a compelling explanation and weakens the paper's core argument. While reviewers acknowledged the simplicity and clarity of the approach, significant concerns remain about the evaluation methodology, scalability to complex tasks, and the novelty of certain experimental findings.

**Reviewer Concerns:**

**Addressed by Rebuttal:**
- The authors clarified the experimental scope, noting results span multiple architectures (transformer, SSM), model families (Qwen, Llama, Mamba), and scales (0.5B-3B) [Reviewers 5D2n, cjtC, t56p]
- Questions about Zebra puzzles, figure clarity, and evaluation details were adequately explained [Reviewer 5D2n]
- The comparison to Quiet-STaR was addressed by highlighting differences in computational efficiency and training paradigm [Reviewer 5D2n]
- Theoretical guidance was provided through a proof sketch for the path-star task [Reviewer cjtC]

Outstanding Concerns:
- The evaluation in Section 4.4 essentially tests whether models perform better on objectives similar to their training objective, which lacks the impacts [Reviewer vA7g]
- The finding that random future insertion outperforms fixed/goal-relevant insertion undermines the core argument about goal-relevant information being beneficial, and no compelling explanation was provided [Reviewer vA7g]
- Scalability to complex reasoning tasks (mathematical problem-solving, sophisticated long-form generation) remains undemonstrated [Reviewers cjtC, t56p]
- The method relies on heuristic rules for future token placement without strong theoretical guidance for general domains [Reviewer t56p]

**Reviewer Scores:**

Reviewer 5D2n: 6. The reviewer may maintain their marginally positive score, as the authors addressed concerns about Quiet-STaR comparison and clarified experimental details.

Reviewer cjtC: 6. The reviewer may retain their score given that theoretical guidance was provided and experimental breadth was clarified, though concerns about scalability to more general tasks persist.

Reviewer vA7g: 2. The reviewer would likely maintain their rejection score, as the fundamental concern about Section 4.4's evaluation methodology (testing on objectives resembling training) was not satisfactorily addressed, and the random-outperforming-fixed result remains inadequately explained.

Reviewer t56p: 4. The reviewer may slightly adjust their score given the additional clarifications about future token placement principles, but core concerns about small-scale experiments and lack of theoretical grounding for general domains remain.

---

### Decision · Program_Chairs · 2026-01-26

Reject